# WHEN NOVICES TEACH BETTER: IMPROVING BEHAVIORAL CLONING WITH LOW-SKILL DATA

## ABSTRACT

Behavioral cloning (BC), which trains models to replicate behavior from offline demonstrations, is a common approach in reinforcement learning. Several prior works argue that BC requires expert demonstrations and performs poorly when trained on low-skill or suboptimal data. We challenge this assumption by showing that, in certain regimes, training on low-skill demonstrations can yield models that outperform those trained on high-skill data. Since expert data is often costly and scarce, while low-skill data is cheaper and more abundant, this finding has important practical implications. To explain the result, we introduce a measure that quantifies the *resilience* of a policy—its ability to maintain reward under random perturbations—and show that resilience aligns with observed performance differences. Building on this insight, we introduce a new skill-based training curricula—structuring the training process according to policy skill levels—and show consistently improved BC performance compared to treating all data uniformly or filtering for experts. We show our findings in a custom synthetic environment and MuJoCo and validate using human data from Chess and Racing, showing consistency across domains.

## 1 INTRODUCTION

Behavioral cloning (BC) is a form of imitation learning that trains models to replicate behaviors from offline demonstrations (Bain and Sammut, 1995). It is widely used in reinforcement learning, particularly when explicit reward signals are unavailable or when online interaction is impractical due to safety or cost concerns (Zare et al., 2024). In some settings, BC can rival or even surpass offline RL methods (Kumar et al., 2022b; Qin et al., 2022). When trained on human data, it produces models that exhibit human-like behavior—a desirable property for AI systems that interact with people. It is commonly believed, however, that BC performs best with expert, high-skill demonstrations (Kurin et al., 2017) and performs poorly with low-skill demonstrations (Kumar et al., 2022c). As a result, practitioners often adopt Filtered BC (Mandlekar et al., 2021), discarding all but the highest-skill demonstrations. While effective when expert data is plentiful, this approach is limiting in practice, where datasets often span a wide range of skill levels and expert demonstrations are costly or scarce.

This paper revisits the common belief that BC works best with only expert data. We demonstrate that in certain types of low-data regimes, training on low-skill demonstrations can actually outperform training on expert data. Since expert data is often expensive and limited while low-skill data is cheaper and more abundant, this finding has significant practical implications. To explain this counterintuitive result, we introduce a central concept—***resilience***, which quantifies the ability of a policy (and the corresponding demonstrations generated by the policy) to withstand errors without severe degradation in performance. Our findings reveal that high-skill demonstrations often enter states where errors are disproportionately costly, whereas low-skill demonstrations more often avoid such states, resulting in more reliable models which suffer fewer errors.

Insights from psychology and education help motivate this perspective. Experts often rely on tacit knowledge and omit intermediate steps, making their behavior harder to model accurately (Nathan et al., 2001). High-skill strategies may operate closer to failure margins, amplifying the consequences of small modeling errors (Simon, 1997), and compounding errors in BC can cause learned policies to drift into unstable states (Ross et al., 2011). A racing game illustrates the point: expert drivers maximize speed and hug the track's edges, achieving the fastest lap times but leaving little margin

for error, while novice drivers stay near the center and avoid fragile states. With unlimited training data, BC can reproduce expert strategies, but with limited data, errors in expert demonstrations often occur in fragile states, leading to catastrophic outcomes, whereas errors in novice data are more easily absorbed. These perspectives suggest that the usefulness of a demonstration depends not only on skill but also on the resilience of the demonstrating policy, motivating our study of whether and when low-skill demonstrations can yield more reliable BC models than high-skill data and whether this leads to useful insights in overall BC training.

More concretely, in this paper we investigate the role and impact of low-skill data in behavioral cloning across three domains. We use expected reward as our measure of skill: policies with higher rewards are considered high-skill, and those with lower rewards are low-skill. We begin with controlled synthetic environments, where we manipulate rewards in a tree-based domain and simulate policies across different skill levels. This setup allows us to systematically vary environment characteristics and directly test how skill interacts with resilience. We then replicate these results across 12 tasks in MuJoCo. We then turn to Chess, where public datasets provide billions of human games across skill levels. Finally, we study Racing, using human demonstrations obtained through collaboration with a simulator platform, allowing us to evaluate whether our findings generalize to a complex real-world setting. Taken together, these three domains provide complementary perspectives that reveal when and why low-skill data can be more useful than expert data.

Our experiments show that low-skill demonstrations are more resilient, BC models trained on the low-skill data achieve higher performance at small budgets, and this effect disappears when high-skill demonstrations are more resilient. These observations hold consistently across all four domains.

Motivated by these results, we also introduce a new method for training effectively with mixed-skill datasets. Common strategies—such as filtering exclusively for expert demonstrations or training uniformly over all data—do not fully leverage the diversity of available data. We instead evaluate skill-based curricula that organize training by demonstrator skill level. Across both synthetic and real-world domains, these curricula consistently improve behavioral cloning performance. While secondary to our main contribution, these results suggest practical strategies for making better use of demonstrations at varying skill levels.

**Contributions.** Overall, this work revisits the common belief that BC works best when trained only on expert data and makes three contributions:

- We provide the counterintuitive empirical finding that in low-data regimes, BC trained on low-skill demonstrations can outperform models trained on expert data, and show this effect in both synthetic and real-world domains

- We introduce resilience as a metric for demonstration datasets, and show that resilience explains when this phenomenon occurs.

- To showcase the usefulness of this result, we apply the standard curriculum-learning framework and demonstrate that leveraging low-skill data leads to consistent training improvements across synthetic and real-world domains.

## 2 RELATED WORK

**RL with Human Demonstrations.** A substantial body of work addresses reinforcement learning using datasets of human demonstrations. Imitation learning trains models to replicate human policies directly from demonstrations (Hussein et al., 2017). The simplest form, Behavioral Cloning (BC), frames the task as supervised learning: given a state, the model predicts the demonstrated action (Bain and Sammut, 1995). BC is widely applied in domains such as autonomous driving (Wang et al., 2022), robotics (Florence et al., 2022), and gaming (Ruoss et al., 2024), since it avoids explicit reward modeling or transition simulation. It can even outperform reward-based offline RL in some settings (Kurenkov and Kolesnikov, 2022; Kumar et al., 2022b; Qin et al., 2022; Mandlekar et al., 2021). Alternatives to BC have also been developed. Offline RL incorporates reward signals during training (Levine et al., 2020), which can improve task performance (Kumar et al., 2022c). Inverse Reinforcement Learning aims to infer the underlying reward function of human behavior (Arora and Doshi, 2021). Finally, online RL approaches can leverage demonstrations, especially when access to a human oracle is available (Wagenmaker and Pacchiano, 2023).

**Leveraging Low-Skill Human Demonstrations.** We study whether low-skill human demonstrations can improve the training of machine learning models. In supervised learning, this idea has been widely explored in crowdsourcing (Howe et al., 2006; Vaughan, 2018). By recruiting laypeople to provide relatively low-skill annotations, large-scale datasets can be built without relying on costly expert input. This strategy has been highly successful, with prominent examples such as ImageNet (Deng et al., 2009) driving major progress in supervised learning.

In contrast, in the context of BC, prior work has shown that training on noisy or suboptimal demonstrations often degrades performance in BC (Fu et al., 2020; Mandlekar et al., 2021; Kanervisto et al., 2020), with follow-up theory reinforcing the view that BC typically requires expert data (Kumar et al., 2022c). To overcome this, extensions to BC have been proposed: filtered BC discards low-skill demonstrations using post-processing classifiers, improving performance in robotics (Chen et al., 2020; Wang et al., 2023),(Chen et al., 2025; Hejna et al., 2025), gaming (Kurin et al., 2017; Ruoss et al., 2024), and language modeling (Wang et al., 2024); weighted BC assigns importance weights to demonstrations (Ghosh et al., 2024); and conditional BC augments states with contextual variables such as modified rewards (Chen et al., 2021b) or goals (Ghosh et al., 2019). Different from this line of work, we focus on investigating whether and when low-skill data might improve BC training.

More broadly, reinforcement learning research has examined training with low-skill or mixed-skill human data, often using BC as a benchmark for evaluating alternative methods (Fu et al., 2020; Mandlekar et al., 2021; Kanervisto et al., 2020; Guss et al., 2019). Other studies have explored training with low-skill non-human data, typically generated by perturbing expert demonstrations with noise or using partially trained models (Brown et al., 2019; Chen et al., 2021a; Zhao et al., 2023a).

**Critical States.** The resilience metric in our work is related to the notion of critical states in BC and RL—the idea that certain regions of the state space are riskier or more important than others. For instance, Kumar et al. (2022a) define critical states as a subset of states from which taking a wrong action is especially costly to recover. Similarly, Zhang et al. (2020) propose an RL approach that involves first deploying a model in a less critical source environment (e.g., a traffic simulator) to learn representations of especially dangerous states, which can then be avoided once deployed in real-world environments. While related, our resilience concept differs in focus: it characterizes the *policy*, whereas the notion of critical states characterizes the *states*.

**Curriculum Learning.** Our approach builds on the idea of structuring training around demonstrations of varying skill levels. This is inspired by curriculum learning (Bengio et al., 2009), a technique modeled after human and animal pedagogy that organizes the training process to improve learning outcomes. Most research in curriculum learning structures training by gradually increasing difficulty, such as progressing from easy to hard examples. Curricula have also been designed around tasks, model architectures, or evaluation metrics (Soviany et al., 2022). Although first applied to supervised learning, curriculum learning has since gained traction in reinforcement learning as well (Narvekar et al., 2020). In our method, we structure the data by demonstrator skill level, beginning training with low-skill demonstrations and progressively incorporating higher-skill demonstrations.

## 3 SETTING AND METHODOLOGY

### 3.1 PROBLEM SETTING

**Markov decision process (MDP).** Behavioral cloning is formulated in the context of an MDP, defined by the tuple $\{\mathcal{S}, \mathcal{A}, \mathcal{P}, \mathcal{R}, \gamma\}$, where $\mathcal{S}$ is the state space, $\mathcal{A}(s)$ is the action space in state $s$, $\mathcal{P}(s, a)$ represents the state transition for taking action $a$ in state $s$, $\mathcal{R}(s, a)$ specifies the reward for taking action $a$ in state $s$, and $\gamma$ is the discounting factor. In the remaining of the paper, our focus is on finite-time MDP and has set $\gamma = 1$, though our discussion is general to cases with $\gamma < 1$.

**Policy and rewards.** A policy is denoted by $\pi(a|s)$, indicating the probability for taking action $a$ in state $s$. The policy reward, the cumulated rewards for deploying policy $\pi$ in the MDP, is defined as $J(\pi) = \mathbb{E}[\sum_t R(s_t^\pi, a_t^\pi)]$, where $s_t^\pi$ is the random variable denoting the state at $t$ if the agent follows policy $\pi$ and $a_t^\pi$ is the random variable denoting the agent action in state $s_t^\pi$ if the agent follows policy $\pi$. The expectation is over the randomness of the state transition and the policy. We can similarly define the value function $V^\pi(s) = \mathbb{E}[\sum_t R(s_t^\pi, a_t^\pi)|s_0 = s]$ as the reward of following policy $\pi$ when starting with state $s$.

**Skill levels.** The skill level of a policy $\pi$ is defined by the expected reward of the policy $J(\pi)$. We focus on situations when there exist policy functions at a range of skill levels. We denote these policies as $\{\pi_1, \pi_2, ..., \pi_K\}$. Without loss of generality, we assume $J(\pi_1) \leq ... \leq J(\pi_K)$. For low-skill (high-skill) data or demonstrations, we refer to the state-action pairs $(s, a)$ generated by the policy with relatively low (high) rewards among the set of all policies. The total set of state-action pairs generated by the policy function $\pi_k$ comprises a dataset $\mathcal{D}_k$.

**Behavioral cloning.** Behavioral cloning learns a policy function using supervised learning: Given demonstrations from $\mathcal{D}$, we train a model $M$ to map each state to its corresponding action.

## 3.2 RESILIENCE

We define the *resilience* measure as the expected reward of a policy when a single action in the rollout (i.e., a trajectory generated by following the policy) is replaced with a random action, while all other actions follow the policy. Intuitively, resilience is a useful metric because BC models trained on finite datasets inevitably make errors, particularly under small data budgets. The resilience metric captures the extent to which a policy remains robust to these errors.

More formally, let $\zeta^\pi = \{(s_t, a_t, r_t)\}|_{t=1,...,T}$ be the rollout of the policy $\pi$ and $(s_t, a_t, r_t)$ denote the (state, action, reward) at step $t$ in the rollout. Also let $V^\pi(s)$ be the value function of state $s$ if the agent takes policy $\pi$ after state $s$, we can define resilience as follows.

$$\text{Res}(\pi) = \mathbb{E}_{\zeta^\pi} \left[ \mathbb{E}_{k \sim U[1,T]} \left[ \sum_{t=1}^{k-1} r_t + \mathbb{E}_{(a,s') \sim (\pi_{rand}, P(\cdot|s_k,a))} \left[ R(s_k, a) + V^\pi(s') \right] \right] \right]$$

This measure requires knowledge of the underlying policy $\pi$, which is often unknown. Below, we define an empirical approximation that uses the demonstration dataset $\mathcal{D}_\pi$ generated by $\pi$.[1]

$$\overline{\text{Res}}(\mathcal{D}_\pi) = \frac{1}{|\mathcal{D}_\pi|} \sum_{\zeta \in \mathcal{D}_\pi} \frac{1}{T} \sum_{k=1}^{T} \left( \sum_{t=1}^{k-1} r_t + \frac{1}{|\mathcal{A}|} \sum_a \sum_{s'} P(s'|s_k, a) \left( R(s_k, a) + V(s') \right) \right) \quad (1)$$

### 3.2.1 CONNECTION BETWEEN RESILIENCE AND BC PERFORMANCE

Let $\hat{\pi}$ be the BC policy trained on demonstrations from $\pi$, and let $\epsilon$ be its per-timestep imitation error (e.g., probability of deviating from the demonstrator action). Using Theorem 2.1 from Ross et al. (2011) and adapting from a 0-1 loss formulation to a more general case, we obtain $J(\hat{\pi}) \approx J(\pi) - \epsilon T^2 C_\pi$, where $C_\pi = J(\pi) - \text{Res}(\pi)$ is the cost of making one error and following $\pi$ afterwards [2]. Thus, we get:

$$J(\hat{\pi}) \approx J(\pi) - \epsilon T^2 \big( J(\pi) - \text{Res}(\pi) \big) = (1 - \epsilon T^2) J(\pi) + \epsilon T^2 \text{Res}(\pi). \quad (2)$$

**Implications on low-skill BC.** Consider low- and high-skill demonstrators $\pi_L, \pi_H$ with BC learners $\hat{\pi}_L, \hat{\pi}_H$. Applying Eq. (2) (under a shared-$\epsilon$ approximation) gives

$$J(\hat{\pi}_L) - J(\hat{\pi}_H) \approx (1 - \epsilon T) \big( J(\pi_L) - J(\pi_H) \big) + \epsilon T \big( \text{Res}(\pi_L) - \text{Res}(\pi_H) \big). \quad (3)$$

Since $\epsilon$ typically decreases as the demonstration budget increases, Eq. (3) predicts two regimes: (i) for high budgets (small $\epsilon$), the skill term dominates and higher-skill demonstrations yield better BC; (ii) for low budgets (large $\epsilon$), more resilient demonstrations can yield better BC even if they are lower-skill. In particular, if $\text{Res}(\pi_L) > \text{Res}(\pi_H)$, then there can exist a sufficiently large $\epsilon T^2$ regime where $J(\hat{\pi}_L) > J(\hat{\pi}_H)$.

---

[1]This approximation requires an value function estimator, which could be obtained by training a value function on $D_\pi$ or leveraging known domain value functions (e.g., Stockfish in Chess)

[2]This approximation assumes that BC errors follow a uniform distribution – that $\hat{\pi}$ takes the correct action with $1 - \epsilon$ probability, and takes uniformly random actions with probability $\epsilon$, aligning with our definition of resilience. In practice, BC errors might not be uniform, and accounting for more realistic BC error in our resilience metric would be an interesting future direction.

### 3.3 METHODOLOGY

In this work, we aim to better understand the effect of skill and resilience on behavioral cloning, and how these insights can be leveraged to improve BC. Specifically, we first examine whether training on low-skill data can outperform training on high-skill data, and whether resilience can explain this phenomenon. We then explore how low-skill data can be leveraged to improve BC in practice.

#### 3.3.1 USING RESILIENCE TO EXAMINE LOW-SKILL TRAINING IN BC

We begin by studying whether there are situations where training on low-skill data yields better models in BC than training on high-skill data, and whether resilience can help explain this phenomenon. We consider domains with policies $\pi_1, \ldots, \pi_K$ at varying skill levels, each generating a corresponding demonstration dataset $\mathcal{D}_1, \ldots, \mathcal{D}_K$. We assume the datasets are given and that the skill level of the generating policy is known. Our goal is to examine whether, under different data sizes, training on a low-skill dataset can yield better performance than training on a higher-skill dataset.

We use the empirical resilience measure in Equation 1 to compute the resilience of demonstrations at different skill levels. In particular, we are interested in ordering resilience scores across skills to identify positive or negative trends between resilience and skill. We denote environments where high-skill agents are more resilient than low-skill agents as **Resilient High-Skill**, and those where high-skill agents are less resilient than low-skill agents as **Fragile High-Skill**.

Given this classification, we then run BC on each dataset $\mathcal{D}_k$, consisting of $B$ demonstrations generated by $\pi_k$. Each dataset is split into training and validation sets. We instantiate a model $M$ and train it to minimize loss on the training data, repeating this process for multiple epochs until validation loss converges. Finally, we evaluate the task performance of the trained models at each budget and skill level.

#### 3.3.2 TRAINING WITH SKILL-BASED CURRICULA

Next, we investigate how to leverage low-skill data in the common setting where a mixed-skill dataset $\mathcal{D} = \{\mathcal{D}_1, \ldots, \mathcal{D}_K\}$ has already been collected. We assume that, given a budget $B$, each dataset $\mathcal{D}_k$ contains $\frac{B}{K}$ demonstrations from $\pi_k$. Our task is to choose a training schedule that maximizes the final model's performance, especially under the common scenario where marginal compute costs are much cheaper than data acquisition costs. We consider three possible schedules: two baselines from prior literature and one proposed curriculum that orders training by skill level. Our goal is not to prove that this proposed schedule is the optimal one, but simply show that it is possible to use low-skill data to improve training scheduling in the first place.

- **Standard BC:** In this method, the model is trained using supervised learning to mimic the full training dataset $\{\mathcal{D}_1, ..., \mathcal{D}_K\}$ until convergence.

- **Filtered BC:** The dataset is first preprocessed to include only the data at the highest skill $\{\mathcal{D}_K\}$. The model is then trained using the same BC methodology on the thresholded dataset.

- **Filtered BC-LSC[3]:** We first train a model on the full dataset $\{\mathcal{D}_1, \ldots, \mathcal{D}_K\}$ until convergence (Standard BC). We then iteratively remove the lowest-skill data and retrain on the remaining datasets, ending with training only on $\mathcal{D}_K$ until convergence (Filtered BC).

To assess the effect of different training curricula, we train a model under each schedule to obtain three models and evaluate their task performance in the domain.

### 3.4 DOMAINS

#### 3.4.1 TREE

We design a simplified class of tree environments to investigate the effect of data skill-level and resilience on BC model performance. [4] Each environment we construct is characterized by a depth-5

---

[3]*LSC* stands for Low-Skill Curriculum

[4]Our experiment code, including both tree and chess domains is available at the following anonymized link: https://anonymous.4open.science/r/iclr2026-C020

binary tree structure, where each node in the tree represents a state. For each non-terminal node, the agent's action space consists of moving to either the left or right child node. The state transitions are deterministic. Each node is associated with a reward, and the agent collects the reward from each node it passes through.

To generate demonstrations at different skill levels, we model an agent's skill based on how many steps it can look ahead. Specifically, an agent with skill level 1 always chooses the direct child node with the larger reward. An agent with skill level $k$ evaluates all possible combinations of the next $k$ moves and selects the next move that leads to the optimal path over those $k$ moves.

In order to better understand the relationship between resilience and BC performance, we specifically design a set of environments where high-skill agents have lower resilience. Specifically, each environment is randomly generated with different rewards assigned to the nodes. For each node, we begin by drawing a reward value uniformly at random between 0 and 1. We then randomly designate a subset of the nodes as *fragile*, with the proportion of fragile nodes varying based on the experiment. For these nodes, we decrease their rewards by 8, and randomly select one of their child nodes to increase its reward by 10. The intuition of this design is to provide a connection between the behavior of low-skill and high-skill agents to our metric of *resilience*.

Using this definition of skill, we observe that low-skill and high-skill agents follow different strategies in an environment. The lowest-skill agent will always avoid fragile nodes if possible, to avoid paying the entry penalty. In contrast, the highest-skill agent will explicitly seek out these states, recognizing that the additional reward obtained from them improves its overall score. However, this behavior implies that high-skill agents are less resilient to fragile nodes than low-skill nodes.

Therefore, environments with a high proportion of fragile nodes can be classified as fragile high-skill, while environments without any fragile nodes can be classified as resilient high-skill. We can tune this parameter to understand the effect of resilience on BC performance. We provide more details about the domain construction, as well as an example tree diagram in Appendix A.1.

### 3.4.2   MuJoCo

MuJoCo is a physics engine used for simulating robotics and biomechanics and contains 11 different environments for training agents. (Todorov et al., 2012). We use an existing set of pretrained models which span several combinations of environment, architecture, and skill (Younis et al., 2024). In our results, we evaluate on every environment/architecture combination with at least two models (except for SAC-HumanoidStandup, which failed to run). We denote the highest skill model as "high-skill" and the lowest-skill model as "low-skill". Overall, this gives us 12 additional tasks to evaluate.

### 3.4.3   CHESS

The previous two examples allow us to examine training behavior in a simplified toy scenario, and a more complex set synthetic environments. Next, we aim to show that these results hold in a real-world setting, and first pick the domain of Chess. Chess is a useful domain to analyze because of the huge human datasets available for analysis, a simple and well-defined MDP, and a highly tuned metric of policy skill. We conduct our experiments using two datasets. First, we use the Lichess Player Database (Lichess, 2025), which contains 6.2 billion human chess games at skill levels ranging from beginners to grandmasters. Second, we construct a dataset of Stockfish games (the best chess engine available), representing skill level as nodes searched. For all experiments in this domain, we use the BC training framework provided by McIlroy-Young et al. (2020a). More details on the model training and Stockfish game generation processes is available in Appendix C.

### 3.4.4   RACING

We also extend our exploration of low-skill BC to the domain of Racing. Through an academic partnership with a commercial racing simulator platform, we obtained access to human driving trajectories from a Formula 4 racing series. The dataset contains 10,000 complete laps on each of four tracks: Summit Point Raceway, Circuito de Navarra, Autodromo Nazionale Monza, and Road America (a diagram of each map is available in Appendix D). We approximate skill by lap time, with the fastest 5,000 laps representing high-skill data and the slowest 5,000 laps representing low-skill data. While our limited access to the platform prevents us from directly computing resilience scores,

our intuition strongly suggests that low-skill demonstrations may be more resilient, due to the nature of high-skill drivers to drive faster and cut corners close to the edge of the track. [5]

# 4 RESULTS

## 4.1 TREE

Following the methodology from Section 3.3.1, we train BC models on datasets generated by agents of each skill level from $k = 1$ to $4$, using data budgets ranging from $2^8$ to $2^{18}$ data points. More details on the BC training process is available in Appendix A.1.

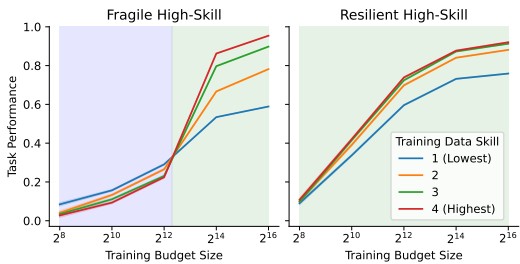

After training, we evaluate each model by simulating 16,384 randomly generated episodes and report the average task performance. We repeat this process with proportion of fragile states ranging from 0 to 1. For each environment configuration, we train models with 200 random seeds and present the mean and standard error.

Figure 1: Effect of training BC models across budgets and skill levels in the tree domain without and with fragile nodes. Regions where low-skill training outperforms high-skill are highlighted in blue.

In particular, we highlight two state configurations - with 20% of tree nodes as fragile and with 0%, to better explore the effects of these states on low-skill BC. We plot the task performance of low-skill and high-skill BC at a variety of budgets for both experiment configurations in Figure 1.

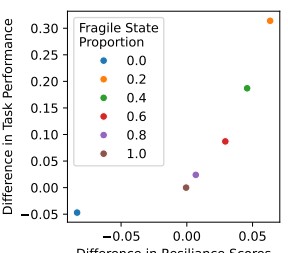

Overall, we see that in the fragile environment configurations, models trained on low-skill data outperform models trained on high-skill data at small budgets (under $2^{14}$), while at larger budgets, training on high-skill data leads to improved task performance. However in the environment without fragile states, training on high-skill demonstrations improves performance at all data budgets. These differences are significant with $p < .001$.

We train each model until convergence, meaning that different models may use different amounts of compute. This is a reasonable assumption in domains where data collection is much more expensive than compute. However, one might ask if low-skill models are using more compute in order to get their higher performance. In Appendix A.1, we show that this is not true, and that low-skill models actually take fewer epochs to converge.

Figure 2: Comparing the difference in resilience and small-budget task performance by tree environment.

To illustrate the potential connection between *resilience* and the benefit of low-skill training, we plot the differences in agent resilience and the difference in final model performance at low budgets ($2^{10}$) across environment configurations. We estimate agent resilience using Equation 1, and plot the difference in resilience and difference in task performance between the lowest and highest-skill agent for each environment configuration in Figure 2.

We see that a clear trend between the difference in resilience scores between the low and high-skill agents and BC model performance. When the low-skill agent has a higher resilience score, the corresponding low-skill BC model has a higher task performance during deployment.

## 4.2 MUJOCO

In each MuJoCo environment, we train BC models on the low-skill agent and the high-skill agent. For both models, we follow Equation 1 and compute the resilience score by rolling out the original policy for up to 1000 steps, and measure the accumulated reward over 5 timesteps after perturbations. We present the results in Table 1. Overall, we see that in 5 domains, the low-skill agent demonstrates higher resilience, and in the other 7 domains, the high-skill agent demonstrates more resilience.

---

[5]We are unable to release the dataset and code for this application due to our agreement with the provider.

Next, we compute the task performance of models trained on low-skill and high-skill data. Following the methodology from Section 3.3.1, we generate datasest consisting of 1000-step rollouts by each agent. We then train models at budgets ranging from $2^2$ to $2^{10}$ until validation convergence. Finally, we evaluate the models by performing 10 rollouts, and reporting the cumulative reward attained. We repeat this procedure for 10 seeds, and present the results in the table above.

In Table 1, we report a summary of results in all 12 domains. The full set of training plots, measuring model performance at multiple training budgets is available in Appendix B.

In all five environments with low-skill resilience, we find that low-skill models indeed outperform high-skill models consistently at low budgets. Additionally, we find that in five of seven domains with high-skill resilience, low-skill models consistently outperform high-skill models at low budgets.

Overall, we find that the presence of low-skill resilience has 100% accuracy in predicting low-skill outperformance. Additionally, when low-skill models are less resilient, they generally fail to outperform high-skill models. The presence of the two exceptions (Pusher and Walker2d) does not invalidate our metric, but suggest that our approximate resilience calculation may not be accurate enough in these domains.

| Domain | High-Skill Resilience | Low-Skill Resilience | Low-skill more resilient? | Low-Skill performs better? | Aligns |
|---|---|---|---|---|---|
| SAC-HalfCheetah | -0.32917 | **-0.211631** | ✓ | ✓ | ✓ |
| SAC-Hopper | 0.864271 | **0.906312** | ✓ | ✓ | ✓ |
| SAC-Humanoid | 4.807381 | **4.846941** | ✓ | ✓ | ✓ |
| TQC-HalfCheetah | -0.242331 | **-0.231321** | ✓ | ✓ | ✓ |
| TQC-Humanoid | 4.827656 | **4.848168** | ✓ | ✓ | ✓ |
| PPO-Swimmer | **0.04955** | 0.044526 | ✗ | ✗ | ✓ |
| SAC-Ant | **-0.06562** | -0.345851 | ✗ | ✗ | ✓ |
| SAC-InvertedDblPend | **7.49** | 7.47 | ✗ | ✗ | ✓ |
| SAC-InvPendulum | **0.678** | 0.674 | ✗ | ✗ | ✓ |
| SAC-Pusher | **-0.41236** | -0.420937 | ✗ | ✓ | ✗ |
| SAC-Reacher | **-0.14926** | -0.149267 | ✗ | ✗ | ✓ |
| SAC-Walker2d | **0.904928** | 0.812948 | ✗ | ✓ | ✗ |

Table 1: Resilience scores and model performance summary across 12 architectures and environments in MuJoCo.

### 4.3 CHESS

We train two sets of models - human-like models, and engine-like models. For our human-like models, we select five skill levels from the Lichess dataset - moves by players rated at 800 (low), 1200, 1600, 2000, and 2400 (high). For our engine-like models, we select three skill levels, nodes=100 (low), nodes=1000, and nodes=10000 (high). Based on experiments conducted by Marco (2021), these three skill levels of Stockfish roughly correspond to human ratings of 1200, 1700, and 2500. In the main paper, we only focus on the highest and lowest skill levels, but show that the results are qualitatively similar for all skill levels in Appendix C.1.

First, we compute the resilience scores of the human and engine agents at each skill level. We sample 4096 actions from the training datasets at each skill level and use Equation 1 with the highest-depth Stockfish as a value function to compute resilience. We plot these scores in Table 2. In Appendix C.2, we show that these resilience scores are robust to weaker approximate value function and at multiple skill levels.

| Skill | Human | Engine |
|---|---|---|
| **Low** | 0.305 | 0.247 |
| **High** | 0.251 | 0.298 |

Table 2: Resilience scores for humans and engines for the lowest and highest skill levels.

We see that in the human dataset, resilience values decrease as skill level increases, with the highest skill level having the least resilience. On the other hand, in the engine dataset, resilience values increase as skill level increases, with the highest skill level having the most resilience. [6] Thus, we denote the environment with human data as Fragile High-Skill, and the environment with engine data as Resilient High-Skill.

To validate our tree-domain results, we again follow the methodology from Section 3.3 and train BC models on datasets containing demonstrations at each skill level, with data budgets ranging from $2^{16}$ to $2^{24}$ data points. We evaluate

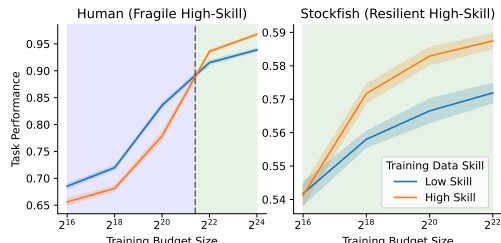

Figure 3: Effect of training BC models on human and engine demonstrations in chess by skill level. Budget regions where low-skill training outperforms high-skill is highlighted in blue.

---

[6] In Appendix C, we show an expanded plot showing this trend is monotonic across more skill levels.

the resulting models by playing 10240 chess games against a randomly-playing opponent. [7] We repeat this process with 20 random seeds and present the mean and standard error for each experiment configuration in Figure 3.

In the human-trained models, we observe that with small data budgets, under $2^{22}$, training on low-skill data achieves the highest performance, while at larger data budgets leads to the best performance. However, with the engine-trained models, we observe that training on the highest-skill data leads to the best task performance at all data budgets. These differences are significant with $p < .001$.

Both results match our intuition and our previous results. In the environment where low-skill agents are more resilient, models trained on low-skill data outperform high-skill models at small budgets, but when this isn't true, high-skill models outperform low-skill models at all budgets.

## 4.4 RACING

Again following methodology from Section 3.3, we train BC models on low and high-skill data on each of the four tracks, at data budgets ranging from 500 to 1000 laps. We evaluate each model using a virtual joystick (pyv, 2025) that forwards the controls to the simulator and measure track completion rate over 5 attempts. We present the results in Table 3.

We see that at the smallest budget, low-skill models complete 60% of races, while high-skill models complete 20% of races. This difference is significant with $p < 0.01$. We see that this trend holds in three out of four tracks. For example, in Summit Point, with a budget of 500 laps, a model trained on low-skill data can always complete the track, whereas a model trained on high-skill data can only do so reliably with 1000 or more laps. The only exception is Road America, where the model trained on high-skill data outperforms even at low budgets. To explain this, we find in Road America, high-skill models qualitatively perform more resiliently, which again aligns with our explanation. We provide more details of this phenomenon in Appendix D.

| Track Name | Budget | 500 | 750 | 1000 |
|---|---|---|---|---|
| Summit Point | Low | 1.0 | 1.0 | 1.0 |
| | High | 0.0 | 0.8 | 1.0 |
| Circuito de Navarra | Low | 0.4 | 0.8 | 1.0 |
| | High | 0.0 | 0.4 | 1.0 |
| Autodromo Nazionale Monza | Low | 1.0 | 1.0 | 1.0 |
| | High | 0.6 | 1.0 | 1.0 |
| Road America | Low | 0.0 | 0.0 | 1.0 |
| | High | 0.2 | 1.0 | 1.0 |

Table 3: Effect of training BC models in a racing simulator by track, budget and skill. In the first three tracks, low-skill training outperforms high-skill training, while in the fourth track, high-skill training always outperforms low-skill training. Budget regions where low-skill training outperforms high-skill is highlighted in blue.

## 5 SKILL-BASED CURRICULA

Next, we conduct experiments to evaluate if applying a standard curriculum schedule over demonstrator skill can improve performance when mixed-skill demonstrations are available.

### 5.1 TREE

We construct a tree dataset with a mixture of demonstrations from agents at four skill levels. We implement both the baselines, Standard BC and Filtered BC, as well as our proposed Filtered BC-LSC approach. We then train models with data budgets ranging from $2^8$ to $2^{20}$ data points. After training, we evaluate each model

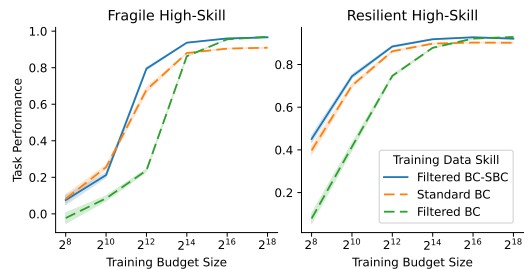

Figure 4: Effect of low-skill curricula on task performance in the tree domain, with environment configurations both containing and not containing fragile states.

by simulating 16,384 random episodes and report the task performance. We repeat this process with 200 random seeds and present the mean and standard error for each configuration in Figure 4.

---

[7]In Appendix C, we replicate our results with other evaluation approaches.

We observe a few key trends. In both environments, we see Standard BC outperform Filtered BC at small budgets (under $2^{14}$), while Filtered BC performs better with larger budgets. This aligns with existing work which suggests Filtered BC is best when budgets are large enough to keep sufficient data after filtering. Second, we see that our proposed approach minimizes the downsides of both approaches at all budgets – Filtered BC-LSC matches or exceeds Standard BC at low budgets, while performing competitively at large budgets against Filtered BC. Overall, we see that simple curriculum strategy leveraging both low-skill and high-skill demonstrations allows us to perform well in both small-budget and large-budget regions.

## 5.2 CHESS

Following the methodology from Section 3.3.2, we train Standard BC and Filtered BC models, with and without a low-skill curriculum, with data budgets ranging from $2^{16}$ to $2^{24}$, using the low- and high-skill agents in both the human and Stockfish datasets. We evaluate the resulting models by playing 10240 chess games against a randomly-playing opponent, and report the average win rate of the model. We repeat this process with 20 random seeds, and present the mean and standard error for each experiment configuration in Figure 5.

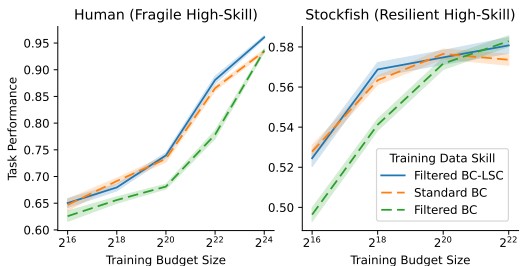

Figure 5: Effect of training BC models on human and engine demonstrations using skill-based curricula in chess by skill level.

Overall, we observe similar trends to those seen with the tree domain results. Standard BC outperforms Filtered BC at lower budgets, while Filtered BC outperforms at higher budgets. Again, we see that Filtered BC-LSC acts as a combination of the two, performing as well as Standard BC at low budgets, while better taking advantage of high-skill data at high budgets. Overall, this gives us strong evidence that it is possible to leverage low-skill data to train BC models with high performance.

## 6 CONCLUSION AND DISCUSSION

We demonstrate that low-skill demonstrations can be leveraged to improve the performance of BC models. Our findings show that BC trained on low-skill data can outperform BC trained on high-skill data in small-data regimes. This effect is strongly correlated with the *resilience* of the training datasets: it arises in domains where low-skill trajectories are more resilient to mistakes, but not in domains where this is not the case. We also show that a simple curriculum learning–like strategy that organizes data by skill level can improve BC performance compared to standard approaches.

These results lead to several useful takeaways. When practitioners have reason to believe that low-skill policies are more resilient, they should consider training on low-skill data. While this behavior is not guaranteed, our findings suggest that practitioners can benefit from testing whether it holds in their specific domain. In scenarios where only small, mixed-skill datasets are available, emphasizing low-skill data during training can be particularly valuable, especially when compute costs are much lower than data acquisition costs. By contrast, in settings with large data budgets, low-skill data is less helpful, and practitioners should instead focus on collecting the highest-quality demonstrations.

There are also several open directions for future work. We showed that a simple curriculum schedule can lead to increased performance, but more complex schedules may show additional benefits, especially developed jointly with online algorithms that dynamically determine which data composition to acquire. Second, methods for automatically estimating the complexity of human strategies in each domain would better guide practitioners in deciding which skill data to collect. Finally, it would be valuable to explore whether the benefits of training on low-skill data extend beyond BC to offline RL or other learning paradigms.

## ETHICS STATEMENT

This work uses human-generated data from two primary sources: a public chess database and a private racing simulator dataset. The chess data is drawn from the Lichess Player Database, which consists of anonymized games that are publicly available for research. The racing data was obtained through an academic partnership with a commercial simulator platform. This dataset was handled in accordance with our data sharing agreement with the provider, and all data was anonymized to protect user privacy. The racing platform was not involved with the research direction or conclusions of this paper, and there are no conflicts of interest between the platform and this work.

Our research aims to create more robust and reliable models through behavioral cloning, and we do not foresee direct negative societal impacts. On the contrary, our finding that low-skill data can improve model resilience may contribute to the development of safer AI systems, as it encourages learning from a wider, more cautious range of behaviors rather than solely from "expert" demonstrations that may operate close to failure margins.

We have used large language models to assist with proofreading and polishing writing of this paper.

## REPRODUCIBILITY STATEMENT

We have made every effort to ensure the reproducibility of our results. Our experiment code for both the synthetic tree and chess domains has been made available in an anonymized repository at https://anonymous.4open.science/r/iclr2026-C020. The core methodology, including the empirical definition of our resilience metric (Equation 1) and the design of our skill-based training curricula, is described in Section 3.3. For our experiments, detailed descriptions of the synthetic tree domain construction, agent generation, and model training procedures are provided in Section 4 and Appendix A. Similarly, implementation details for the chess experiments, including data sources, model architecture, and evaluation protocol, can be found in Section 5 and Appendix B. While the proprietary nature of the racing dataset used in Section 6 prevents its public release, we have detailed the experimental setup and provided qualitative analyses in Appendix C. We believe these resources provide the necessary details to reproduce our key findings. .

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

## A    TREE EXPERIMENTS

### A.1    DOMAIN CONSTRUCTION

As described in Section 4, each tree task is composed of a binary tree structure. Each node in the tree is randomly assigned a reward between 0 and 1. We also designate a subset of the nodes as *fragile*. These nodes have an extra penalty for reaching them (-8), but an even larger reward (10) after making the correct action in this state. However, if the wrong action is taken in this state, no extra reward is given, and the agent simply suffers the initial penalty. An example initialization of a tree structure is visualized in Figure 6, with an example of a fragile state and extra-reward state highlighted in orange and green, respectively.

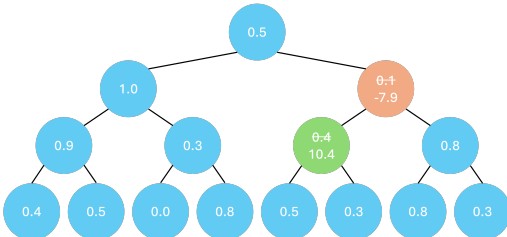

Figure 6: An example tree structure, with a fragile state highlighted in orange, and the extra reward state highlighted in green.

With this definition of state fragility, we can then characterize entire tree domains by the proportion of states which are randomly assigned as fragile, with this proportion ranging from 0 to 1.

### A.2    MODEL TRAINING

In Section 4.1 we conduct all experiments with a tree depth of 5. To train our BC models, we construct datasets which contain data points on the action an agent would take at a randomly generated state. We then create multiple datasets at budgets ranging from $2^8$ to $2^{16}$, at skill levels ranging from $k = 1$ to 4, with fragility proportions in the set $\{0, 0.2, 0.4, 0.6, 0.8, 1.0\}$.

On each generated model, we train a BC model. This model consists of a simple fully-connected neural network with 1 layer, and 16 nodes. We allocated 75% of the dataset for training, and the rest for validation. We trained the neural network on the training dataset using the ADAM optimizer with the initial learning rate of 0.1, and trained until validation loss converged (measured using a patience level of 5 epochs). The full training code is available at https://anonymous.4open.science/r/iclr2026-C020.

While training until convergence is a reasonable assumption in domains with expensive data acquisition costs, one question may arise – are we spending too much in compute costs to get improved performance at the low-skill, low-budget region? In Table 4, we find that this is not true – in fact, low-skill models take fewer epochs to converge than high-skill models at the low-budget region.

| Budget | Low-Skill Epochs | High-Skill Epochs |
|:---:|:---:|:---:|
| $2^8$ | 223.6 | 246.0 |
| $2^{10}$ | 161.9 | 189.2 |
| $2^{12}$ | 821.6 | 846.5 |
| $2^{14}$ | 1375.6 | 635.6 |
| $2^{16}$ | 1020.7 | 470.3 |

Table 4: Comparison of Low-Skill vs. High-Skill Epochs across budgets.

For each generated model, we evaluate its task performance by generating an additional set of $2^{14}$ tree environments using the same configuration that the model was originally trained on, and evaluate the expected performance of trained agent in these environments, by computing the average sum of rewards encountered by the agent.

The full set of results is available in Figure 7

In Section 4.2, we largely followed the same training paradigm as in Section 4.1, with two differences.

(1) First, we constructed training datasets which contained a mixture of actions across skill levels. For instance, with Standard BC with a budget of $2^{10}$, the dataset actually contained $2^8$ data points at $k = 1$, $2^8$ data points at $k = 2$, $2^8$ data points at $k = 3$, and $2^8$ data points at $k = 4$, split into 75% training and 25% validation as before.

(2) We construct a training curricula which consists of a series of training datasets. In each step, we train the model until validation convergence. In the next step, we take the best model from the previous step, and re-run training on the next dataset, after entirely resetting the learning rate and optimizer state.

## B  MUJOCO EXPERIMENTS

In Section 4.2, we replicated our experiments from Tree in a more complex set of environments. In Table 1, we printed a summary of the results of low-skill and high-skill training. Below, we provide a more comprehensive set of results showing task performance for each environment, demonstrator skill, and training budget.

## C  CHESS EXPERIMENTS

In Section 4.3, we repeat our experiments using Chess instead of a synthetic domain. Broadly, we follow the experimental decisions taken by McIlroy (McIlroy-Young et al., 2020b), which was itself inspired by the training process of the Leela chess engine. We collect human trajectories from the online platform Lichess.org. Each action is labeled according to the skill level of the human who made it, where 400 is the lowest possible skill, and 3400 is roughly the highest skill (though technically the range is uncapped).

In addition to the human data, we construct a synthetic demonstration dataset using the leading chess engine: Stockfish. We have Stockfish play matches against itself, varying the node depth of Stockfish search to simulate skill levels.

### C.1  MODEL TRAINING

Each state in the training dataset is represented as a $112 \times 8 \times 8$ matrix, and each action is represented as a one-hot encoded vector of length $1858$. Using the transformer-based architecture of the Leela T74 architecture, we randomly initialize the weights of the model before training the BC model on the human or Stockfish data until convergence using SGD with an initial learning rate of 0.1.

Once the model is trained, we evaluate its task performance by having the trained model play against a completely random (i.e untrained) chess model until the end of the game. Our task score is computed as the average result over a match of independent 10240 chess games.

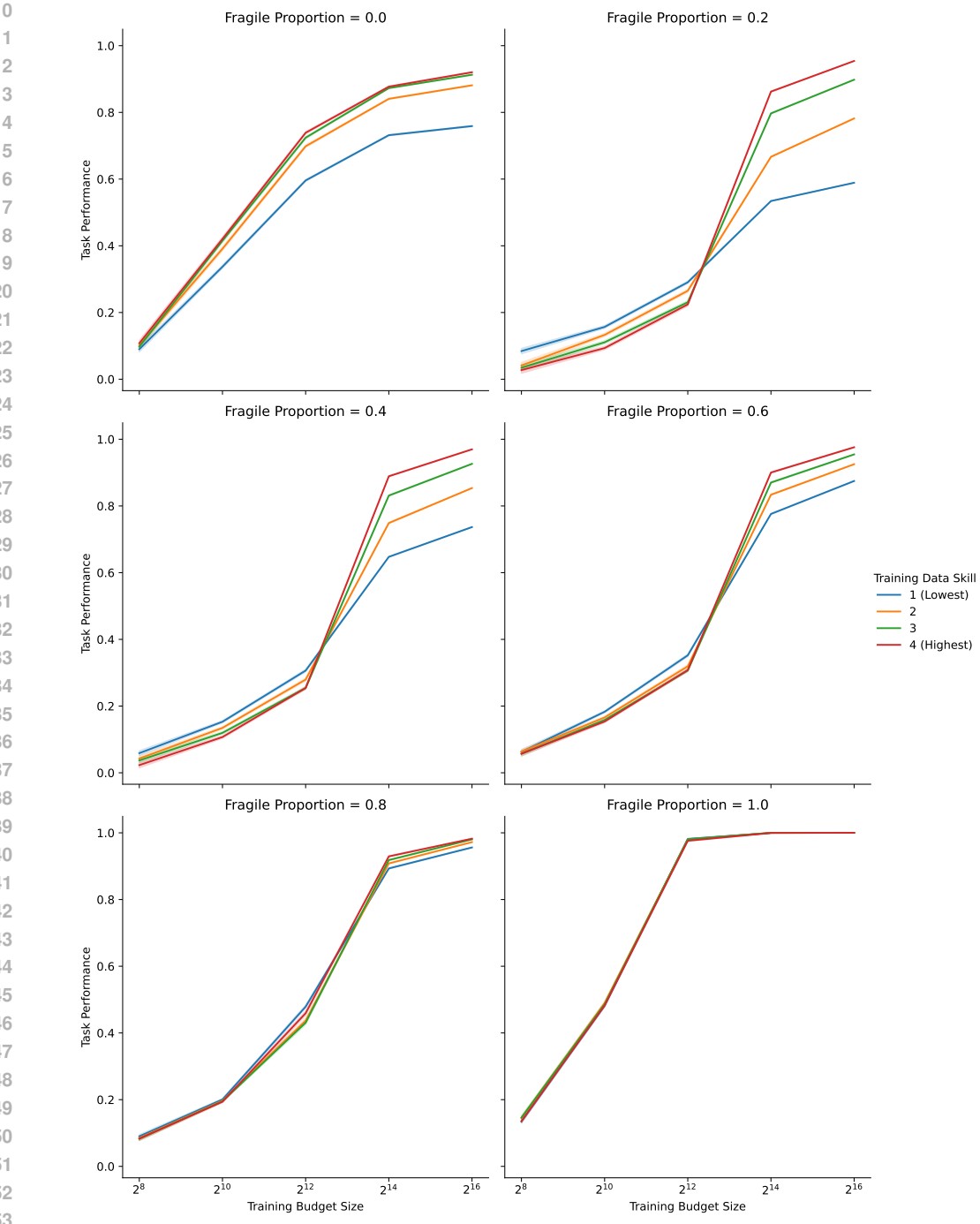

Figure 7: Effect of training BC models across budgets and skill levels in the tree domain at varying proportions of fragile states

The full set of results is available in Figure 9.

In addition to evaluating the model against a completely random opponent, we also evaluate two other approaches, playing against a 800-rated opponent (a model trained to convergence on 256M chess positions at the 800 level), and evaluate the model on a set of puzzles. We present these results in Figure 10 and 11.

| Stockfish Depth | Low-Skill Resilience | High-Skill Resilience |
|:---:|:---:|:---:|
| 1 | 0.31235 | 0.27070 |
| 10 | 0.31217 | 0.27074 |
| 100 | 0.30161 | 0.25938 |
| 1000 | 0.29605 | 0.24837 |
| 10000 | 0.30086 | 0.25364 |
| 100000 | 0.30203 | 0.25337 |

Table 5: Resilience scores at varying Stockfish depths.

## C.2 RESILIENCE COMPUTATION

In Table 2, we showed resilience scores for low-skill and high-skill humans, evaluated using the highest depth stockfish. In Table 6, we show the resilience against other skill agents, and show that these trends still hold, where humans have higher resilience at lower skills, while engines have higher resilience at higher skills. In Table 5, we also show resilience scores for low-skill and high-skill humans, evaluated using multiple depths of stockfish, and we find that the trends hold at all depths.

## D RACING EXPERIMENTS

### D.1 MODEL TRAINING

Through an academic partnership with a commercial racing simulator platform, we obtained access to human driving trajectories from a Formula 4 racing series. Specifically, we use a dataset of 10000 complete laps on each of four different tracks: Summit Point Raceway, Road America, Circuito de Navarra, and Autodromo Nazionale Monza. A reference of these map diagrams is available in Figure 12.

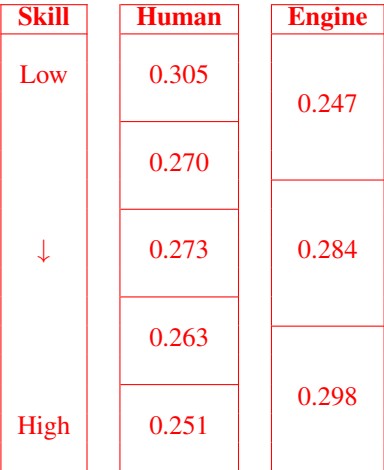

| Skill | Human | Engine |
|:---:|:---:|:---:|
| Low | 0.305 | 0.247 |
| | 0.270 | |
| ↓ | 0.273 | 0.284 |
| | 0.263 | |
| High | 0.251 | 0.298 |

Table 6: Resilience scores for humans and engines across skill levels

We approximate skill by ordering the laps by time, with the fastest 5000 laps representing high-skill data and the slowest 5000 laps representing low-skill data. We evaluate our models using a virtual joystick (pyv, 2025) that forwards the controls to the iRacing simulator.

We train a deep MLP model on state variables (e.g., car speed, heading, etc.) to predict four driving controls (steering, throttle, brake, and gear) at 60Hz. We use action chunking (Zhao et al., 2023b) to ensure stability and accommodate inference latency, predicting 16 future actions at a time.

### D.2 ADDITIONAL ANALYSIS

Figure 13 shows the failure experienced by the low-skill model on Road America. As the trajectory shows, the low-skill model makes a tighter turn and eases off the throttle too late, pushing it to the outer boundary of the track and resulting in a more fragile state. In contrast, the high-skill model takes a wider turn and eases the throttle early (dropping to 0.4), allowing it to complete the turn safely. Thus in this situation, the high-skill model is actually more resilient than the low-skill model, explaining why it performs better even at lower data budgets.

We also show a more typical example in Figure 14 where the low-skill model is more resilient than the high-skill model, in this case at the failure experienced by the high-skill models on Summit Point. Here, the high-skill model is too aggressive on the entry to the turn and does not brake enough (0.6) compared to the low-skill model (0.8).

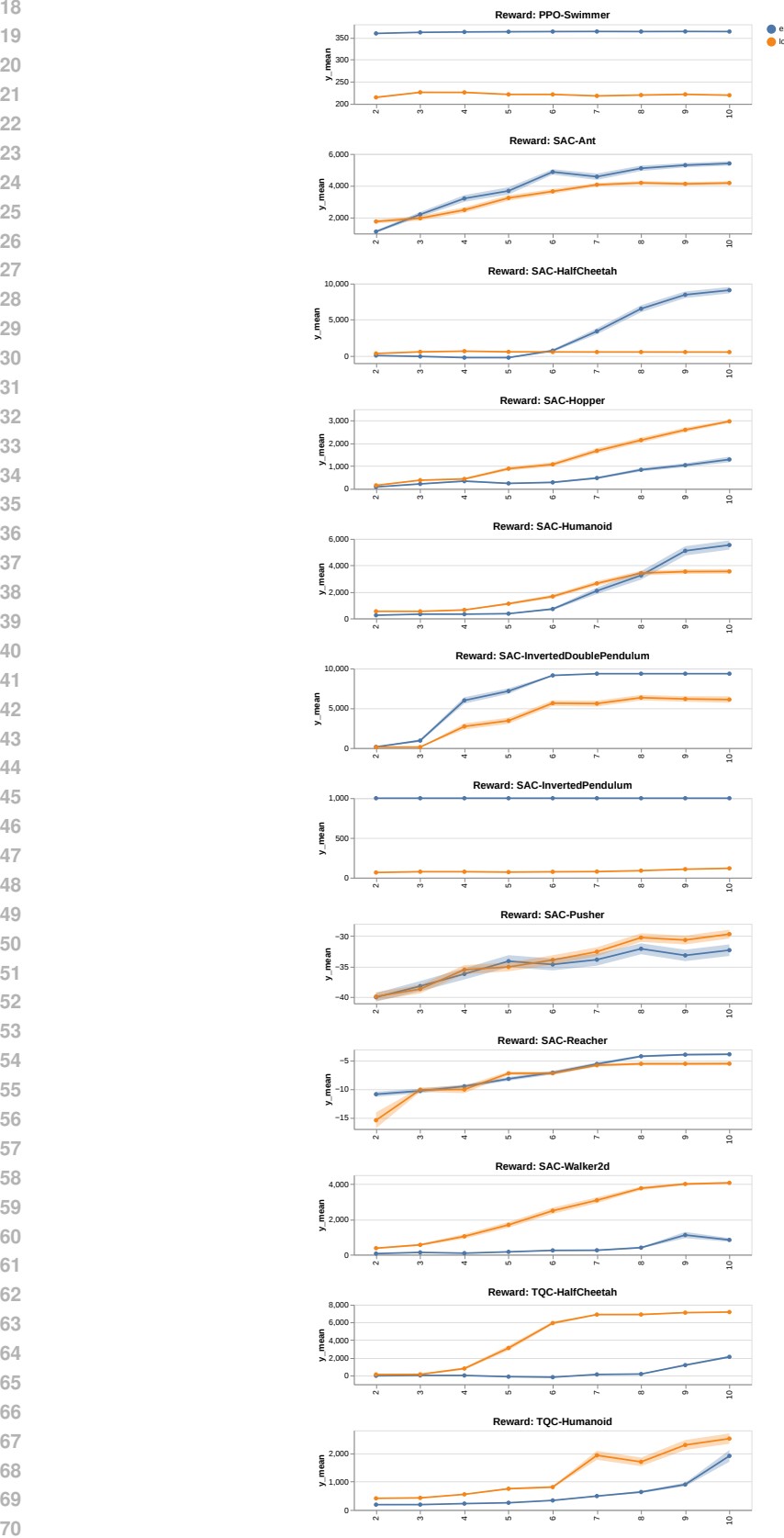

Figure 8: Effect of training BC models across budgets and skill levels in MuJoCo

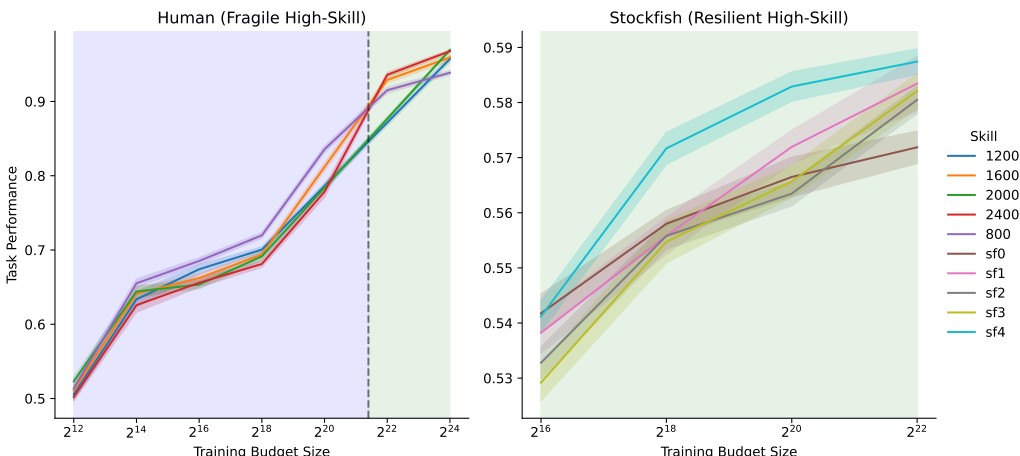

Figure 9: Effect of training BC models on human and engine demonstrations in chess by skill level. Budget regions where low-skill training outperforms high-skill is highlighted in blue.

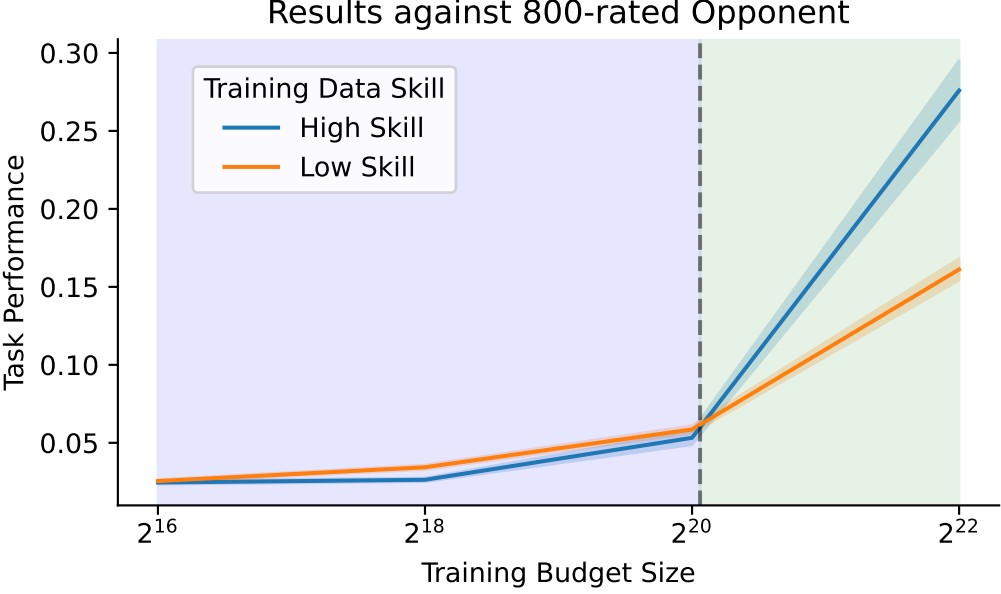

Figure 10: Effect of training BC models on human and engine demonstrations in chess by skill level, evaluating against an 800-rated opponent. Budget regions where low-skill training outperforms high-skill is highlighted in blue.

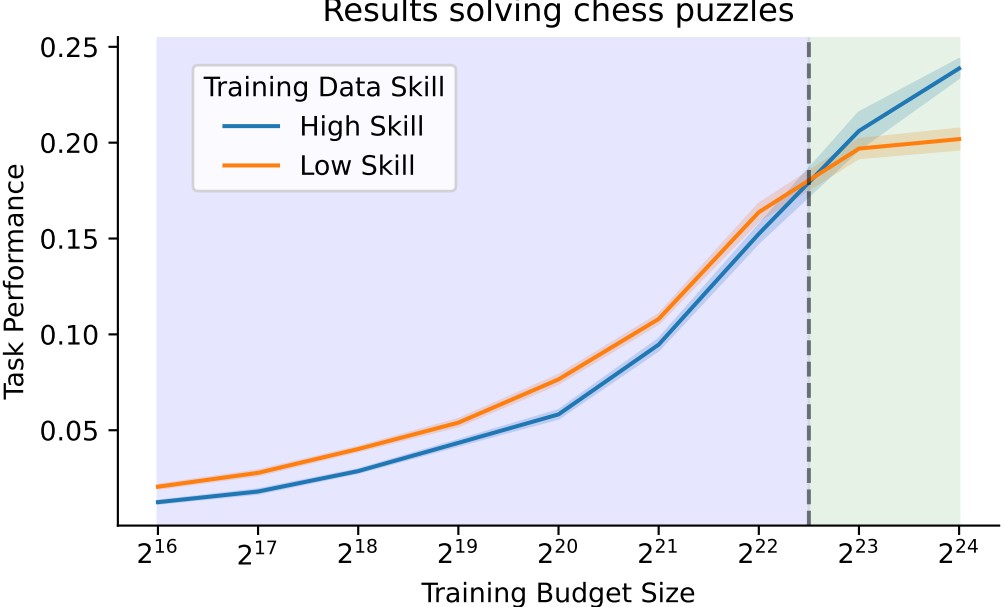

Figure 11: Effect of training BC models on human and engine demonstrations in chess by skill level, evaluated using puzzle solve rate. Budget regions where low-skill training outperforms high-skill is highlighted in blue.

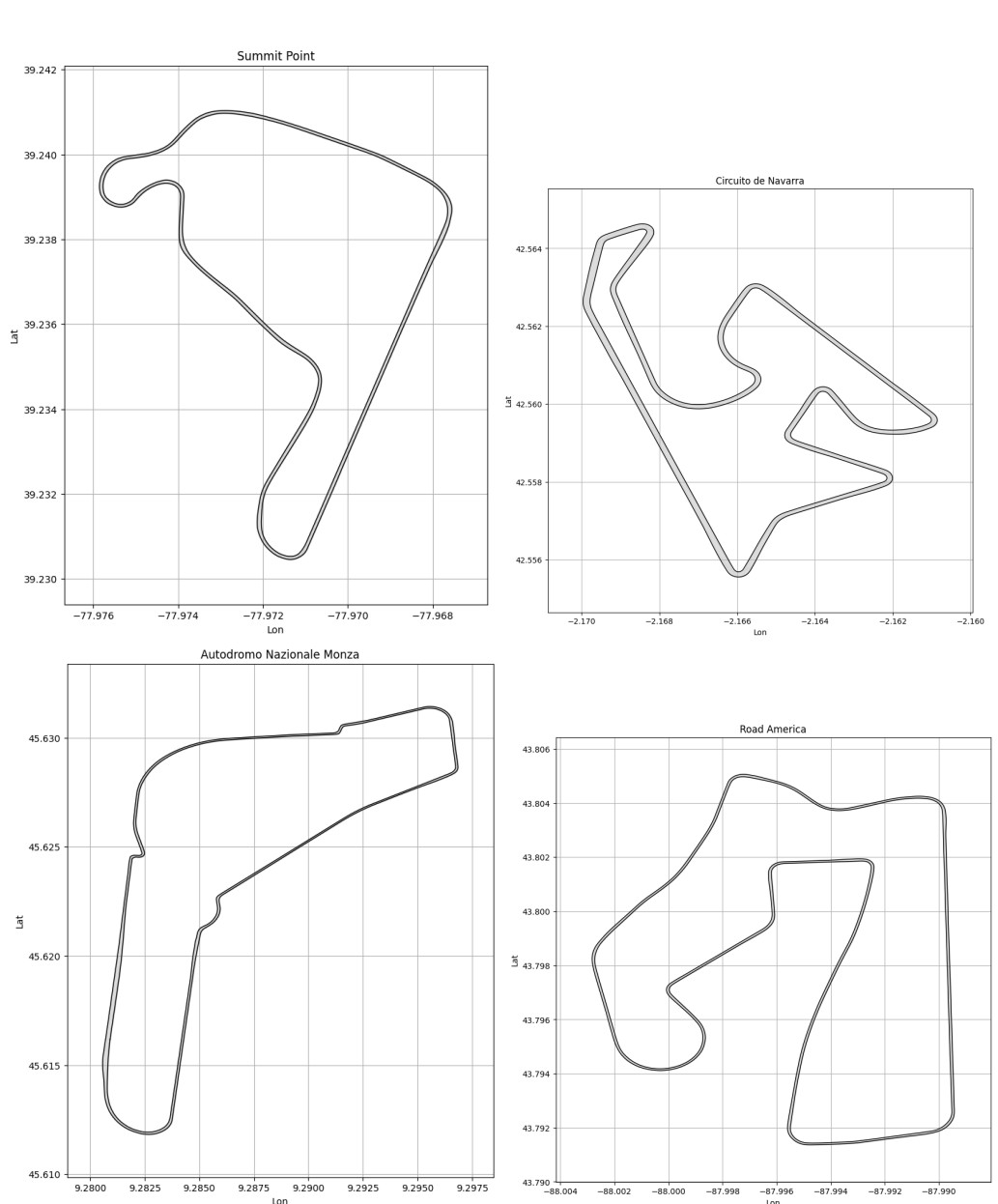

Figure 12: Diagrams of each of the four tracks we evaluate BC on.

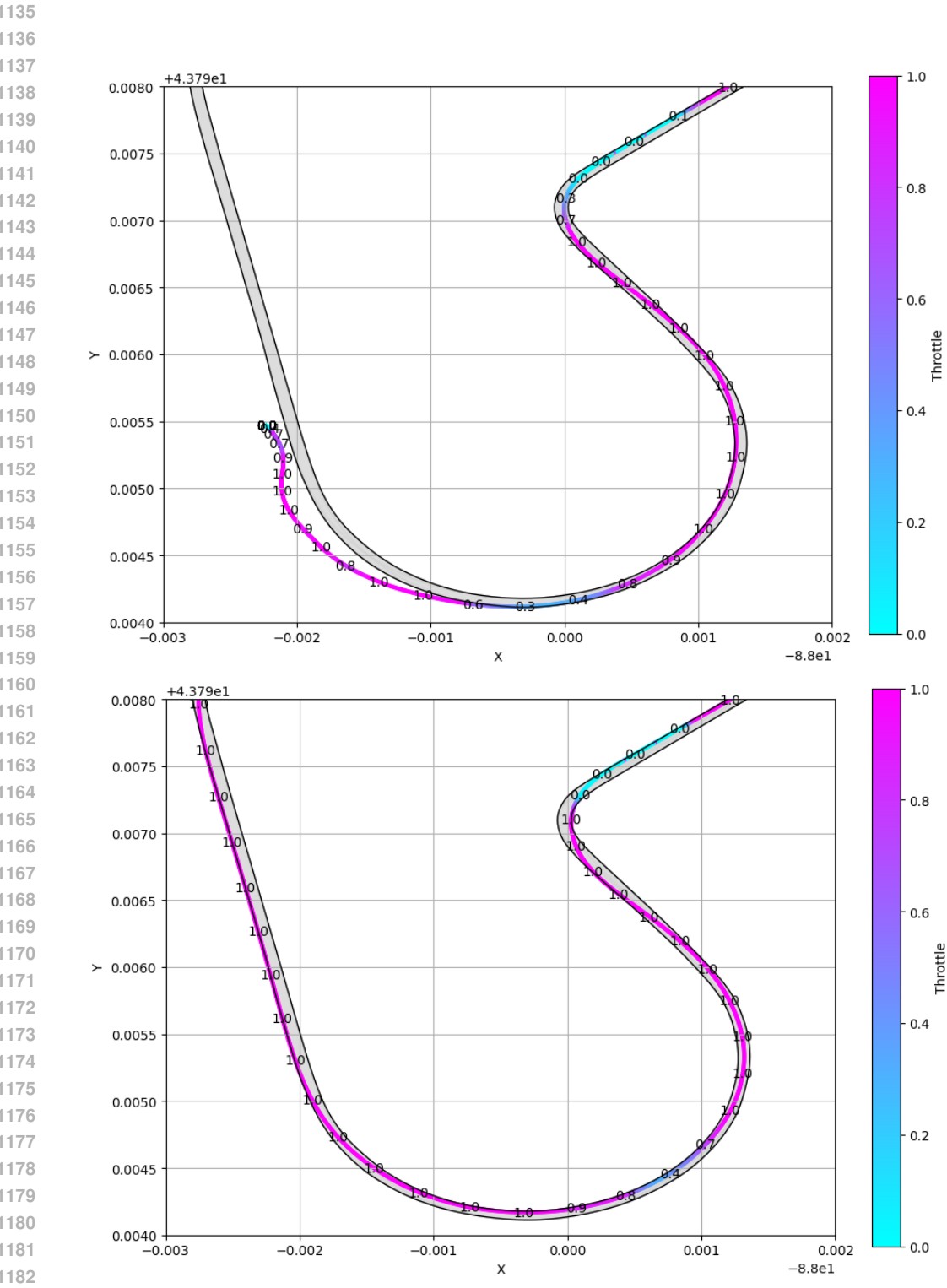

Figure 13: Failed low-skill trajectory and successful high-skill trajectory on Road America

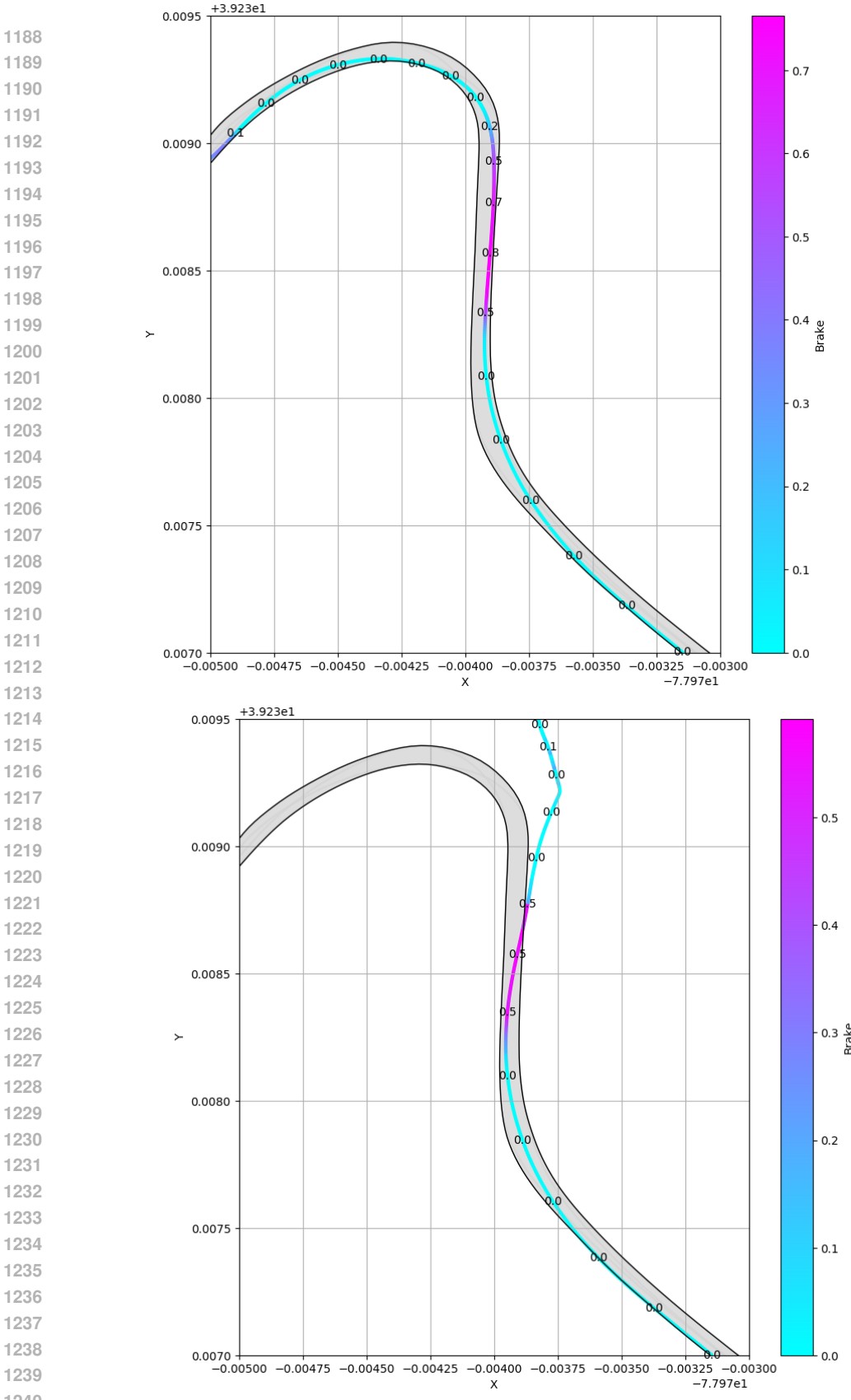

Figure 14: Successful low-skill trajectory and failed high-skill trajectory on Summit Point

