# OpenReview forum: "When Novices Teach Better: Improving Behavioral Cloning with Low-Skill Data"
_ICLR.cc/2026/Conference — Submitted to ICLR 2026_

### Official Review · Reviewer_edUZ · 2025-10-31

**Soundness:** 2
**Presentation:** 3
**Contribution:** 2
**Rating:** 4
**Confidence:** 3

**Summary:**

In this paper, the authors show that in low-data regimes, training BC models on low-skill demonstrations can outperform training on high-skill demonstrations. To explain this phenomenon, they introduce a resilience metric that quantifies a policy's ability to maintain reward under random perturbations. They test their findings across 3 domains: a synthetic tree environment where they control for fragility of states, chess using both human games from Lichess and synthetic Stockfish games, and racing using human driving data from a simulator. The results show that when low-skill policies are more resilient than high-skill policies, BC models trained on low-skill data perform better at small budgets. And they also propose a skill-based curriculum that trains from low-skill to high-skill data.

**Strengths:**

I liked the core observation from this submission that low-skill data can outperform expert data in low-data regimes challenges a fundamental assumption in BC. This is counterintuitive and practically important.

**Weaknesses:**

1. I feel that the resilience metric is computed after observing the phenomenon rather than being used to predict when low-skill data will help (seems quite ad-hoc). The authors designed the tree domain to have fragile states, computed resilience in chess after seeing the results, and didn't even compute it for racing. For this to be a methodological contribution, I need to be able to use resilience on a new domain to make decisions about data collection.
2. There's no formal analysis of when this phenomenon occurs, sample complexity comparisons, or theoretical justification for the resilience-performance connection. Even a simple toy analysis would strengthen the claims considerably.
3. I found no statistical significance testing, confidence intervals, or error bars in many places. Given these are toy datasets, I would expect the authors to run some statistical tests before reporting results and drawing conclusions over them.
4. I feel that the curriculum learning results are weak. Figs. 3 & 5 show Filtered BC-LSC mostly just averaging between the baselines. (Plus there are no error bars to meaningfully conclude something)
5. The authors claim low-skill data is "cheaper and more abundant" but test equal-sized datasets. The real question should be: is it better to get 100 expert demos or 1000 novice demos for the same (or maybe even lesser) cost? Or an ablation where you show the performance comparison for X expert demos vs n*X low-skilled demos.

**Questions:**

1. Can you demonstrate that the resilience metric can predict, on a held-out domain you haven't tested yet, whether low-skill or high-skill data will perform better? This would make the contribution much stronger.
2. At what budget does high-skill start to dominate? Is there a principled way to determine this threshold based on environment properties?
3. Do you have any theoretical or empirical analysis of the sample complexity tradeoff? How many low-skill demos equal one high-skill demo?
4. Typos and Grammatical Errors:
Line 022: "show consistently improve" → grammatical error
Line 076: "for train effectively" → missing "to" ("to train effectively")

---

> ### Author Response · Authors · 2025-11-28
> **Response to Reviewer edUZ (1/2)**
>
> We thank the reviewer for their thoughtful feedback. We are encouraged that they our core observational counterproductive and important, and that they agree that our work challenges a common assumption in BC. In our general response, we have provided details results in an additional experimental setting and provided theoretical insights for our resilience metric. Below we address each of the reviewers specific weaknesses/questions:
>
> > I feel that the resilience metric is computed after observing the phenomenon rather than being used to predict when low-skill data will help (seems quite ad-hoc). The authors designed the tree domain to have fragile states, computed resilience in chess after seeing the results, and didn't even compute it for racing. For this to be a methodological contribution, I need to be able to use resilience on a new domain to make decisions about data collection.
> >
> > Can you demonstrate that the resilience metric can predict, on a held-out domain you haven't tested yet, whether low-skill or high-skill data will perform better? This would make the contribution much stronger.
>
> In response to reviewer feedback, we have replicated our main set of results in MuJoCo. Please see the general response to all reviewers for more details.
>
> > There's no formal analysis of when this phenomenon occurs, sample complexity comparisons, or theoretical justification for the resilience-performance connection. Even a simple toy analysis would strengthen the claims considerably.
> >
> > At what budget does high-skill start to dominate? Is there a principled way to determine this threshold based on environment properties?
>
> We have provided some theoretical justification for the resilience-performance connection, as well as a brief analysis on the high-skill threshold. Please see the general response to all reviewers for more details.
>
> > I found no statistical significance testing, confidence intervals, or error bars in many places. Given these are toy datasets, I would expect the authors to run some statistical tests before reporting results and drawing conclusions over them.
>
> We apologize for the confusion. As noted in the paper (Lines 301, 332, 384, 412), we do report standard error bars for all tree-domain and chess-domain results. These error bars are extremely small, often visually indistinguishable from the mean curves. All of the corresponding comparisons in these two domains are statistically significant. We will make this point clear.
>
> In the racing domain, our evaluation was more limited because each model could only be tested over five rollouts per track. While this restricts our ability to display error bars in the same way as in the tree and chess domains, we can still assess statistical significance across tracks. At the smallest data budget, aggregating the reported results (5 attempts x 4 tracks), low-skill models succeed in 12/20 trials compared to 4/20 for high-skill models—a difference that is statistically significant (p < 0.01). We will include this analysis.

---

> ### Author Response · Authors · 2025-11-28
> **Response to Reviewer edUZ (2/2)**
>
> (continued from previous response)
>
> > I feel that the curriculum learning results are weak. Figs. 3 & 5 show Filtered BC-LSC mostly just averaging between the baselines. (Plus there are no error bars to meaningfully conclude something)
>
> To clarify, the skill-based curriculum is not intended to be a core methodological contribution of the paper. Our main result is the counter-intuitive empirical finding that, in low-data regimes, BC trained on low-skill demonstrations can outperform BC trained on high-skill data, and that this effect is strongly predicted by our resilience metric. The curriculum experiments serve mainly to show that even a simple, off-the-shelf schedule, ordering data by demonstrator skill, can already make use of this insight and improve performance over standard or filtered BC.
>
> We are encouraged that you believe that our approach is hovering around the baselines, and in fact, this was the main conclusion we wanted readers to draw - our approach performs at least as well as Standard BC at small data budgets and performs as least as well as Filtered BC at large data budgets, without having to tune this manually. In addition, there are several points where our approach statistically significantly outperforms both baseline approaches in both domains. We note again that each of these plots contains standard error bars.
>
> > The authors claim low-skill data is "cheaper and more abundant" but test equal-sized datasets. The real question should be: is it better to get 100 expert demos or 1000 novice demos for the same (or maybe even lesser) cost? Or an ablation where you show the performance comparison for X expert demos vs n*X low-skilled demos.
> >
> > Do you have any theoretical or empirical analysis of the sample complexity tradeoff? How many low-skill demos equal one high-skill demo?
>
> Across all our domains, we haven't found a situation where training on more data lowers performance. If having 100 novice demos already outperforms 100 expert demos, then adding cheaper demos will only make these results stronger.
>
> With that being said, this comment highlights a very important aspect of our results, and we perhaps understated the significance of the low-skill vs high-skill performance difference, as the difference may be even larger in practice than the results we showed in this paper. We will highlight this in the paper.
>
> > Typos and Grammatical Errors: Line 022: "show consistently improve" → grammatical error Line 076: "for train effectively" → missing "to" ("to train effectively")
>
> Thank you for the feedback, we will fix these in the text.

---

### Official Review · Reviewer_Y4Mt · 2025-10-31

**Soundness:** 3
**Presentation:** 3
**Contribution:** 2
**Rating:** 4
**Confidence:** 4

**Summary:**

The paper studies when training behavioral cloning (BC) purely on low-skill demonstrations can outperform training on high-skill demonstrations, especially at small data budgets. The authors introduce resilience: the expected return of a policy if exactly one action in a rollout is replaced by a random action (a one-step deviation), after which the policy resumes. They show that when low-skill data is more resilient than high-skill data, BC trained on low-skill can win at small budgets; with enough data, high-skill tends to retake the lead. They also propose a simple skill-ordered training schedule (Filtered BC-LSC): train on all skill levels, then progressively drop lower-skill slices and finish on experts.

**Strengths:**

- Clear, cross-domain empirical pattern: small-budget BC can benefit more from mistake-tolerant (high-resilience) trajectories than from brittle expert ones. The resilience metric offers a practical signal for which skill bucket to prioritize.
- Practical scheduling idea (Filtered BC-LSC) that operationalizes "start broad, finish sharp", consistent with the resilience story.
-  Separation of “which data” from "how much": matched-budget low-only vs. high-only comparisons isolate composition/resilience from sample size; fixed acquisition-budget analyses make the filtering trade-off explicit

**Weaknesses:**

- Narrow perturbation model for resilience: exactly one uniformly timed one-step deviation. BC errors can be state-dependent or clustered; sensitivity to non-uniform or multi-deviation (k>1) settings isn’t explored.
- Oracle dependence: resilience relies on an approximate value function (e.g., a chess engine). Robustness of the resilience ordering to the oracle choice and settings isn’t established.
- Racing details are partly indirect: constraints limit direct resilience computation; some conclusions rely on proxies and qualitative alignment rather than a fully shared evaluator.
- Training-compute accounting for curricula isn’t fully disentangled from the schedule itself (e.g., equalizing optimizer steps across schedules).

**Questions:**

- Composition vs. quantity: can you centralize, in the main text, the matched-budget low-only vs. high-only results and the fixed acquisition-budget schedule results to make the separation between “which data” and “how much data” unmistakable? Any sensitivity to alternative budget splits across skill bins?

- Oracle sensitivity: how stable is the resilience ranking under different value oracles (depths/engines) and reward scalings? A rank-correlation study would help.

- Perturbation model: what happens if the random action is sampled non-uniformly in time, state-dependently, or if you allow multiple random actions (k one-step deviations)? Do the main conclusions persist?

- Continuous control: for racing, how exactly are random actions sampled, and what proxies were used where direct resilience was infeasible? Please document the evaluator and sampling range.

- Compute-matched curricula: do curriculum gains remain when training steps/epochs are matched to the strongest baseline? If so, please add those numbers.


- Mixture selection: with a fixed budget and K skill bins, what mixed proportion of low/high skill maximizes performance? Even a small grid over mixtures in the synthetic domain would be informative.

- Distribution shift: do the resilience performance links hold under test-time shifts (e.g., different chess openings or track conditions)?

---

> ### Author Response · Authors · 2025-11-28
> **Response to Reviewer Y4Mt (1/3)**
>
> We thank the reviewer for their thoughtful feedback. We are encouraged that they find both our proposed approaches (training on low-skill data, training with low-skill curricula) are practical across multiple domains. In our general response, we have provided details results in an additional experimental setting and provided theoretical insights for our resilience metric. Below we address each of the reviewers specific weaknesses/questions below:
>
> > Narrow perturbation model for resilience: exactly one uniformly timed one-step deviation. BC errors can be state-dependent or clustered; sensitivity to non-uniform or multi-deviation (k>1) settings isn't explored.
> >
> > Perturbation model: what happens if the random action is sampled non-uniformly in time, state-dependently, or if you allow multiple random actions (k one-step deviations)? Do the main conclusions persist?
>
> Our goal in introducing resilience was to capture a policy's robustness to random perturbations, which aims to reflect how *BC errors propagate into reward degradation* (see more discussion in the general response)*.* Even though our perturbation model is intentionally simple (one uniformly sampled one-step deviation), it already provides a remarkably reliable signal: across the tree domain (Fig. 1–2), chess (Fig. 4), racing (Table 2), and the new MuJoCo domain, the relative ordering of resilience scores accurately predicts whether low-skill or high-skill training will perform better in low-data regimes. We believe this strong empirical alignment demonstrates the fundamental value of the resilience metric.
>
> That said, we agree that richer perturbation models (e.g., non-uniform timing, state-dependent noise, or multiple deviations k>1) are compelling directions for future work. As the reviewer notes, BC errors may indeed be clustered or state-dependent. Our choice of a simple and easily computable perturbation model was intentional: it allows practitioners to estimate resilience directly from offline demonstrations without introducing additional hyperparameters or requiring environment interaction (a practical constraint in settings such as chess and racing). Exploring more sophisticated perturbation models, while ensuring they remain tractable and interpretable, would meaningfully deepen the theoretical understanding, and we will highlight this as an exciting avenue for future research.
>
> > Oracle dependence: resilience relies on an approximate value function (e.g., a chess engine). Robustness of the resilience ordering to the oracle choice and settings isn't established.
> >
> > Oracle sensitivity: how stable is the resilience ranking under different value oracles (depths/engines) and reward scalings? A rank-correlation study would help.
>
> This is a great point. In Table 1 of the paper, we computed resilience scores using Stockfish with a depth of 100k moves. To demonstrate how sensitive our resilience metric is to different approximation of value functions, below we present results using Stockfish at lower depths. Overall, we find that that the results are qualitatively the same.
>
> | Stockfish Depth  | Low-Skill Resilience |  High-Skill Resilience |
> | - | - | - |
> | 1 | 0.31235 | 0.27070|
> | 10 | 0.31217 | 0.27074 |
> | 100 | 0.30161 | 0.25938 |
> | 1000 | 0.29605| 0.24837 |
> | 10000 | 0.30086 | 0.25364 |
> | 100000 | 0.30203 | 0.25337 |
>
> > Racing details are partly indirect: constraints limit direct resilience computation; some conclusions rely on proxies and qualitative alignment rather than a fully shared evaluator.
>
> Due to the limitations of our access to the training and simulator, our racing results are relatively limited compared with our other domains. To examine the generalizability of our findings, we have conducted additional experiments on MuJoCo. Please see the general responses for the additional results.

---

> ### Author Response · Authors · 2025-11-28
> **Response to Reviewer Y4Mt (2/3)**
>
> (continued from previous response)
>
> > Training-compute accounting for curricula isn't fully disentangled from the schedule itself (e.g., equalizing optimizer steps across schedules).
> >
> > Compute-matched curricula: do curriculum gains remain when training steps/epochs are matched to the strongest baseline? If so, please add those numbers.
>
> As noted on Line 235, we are primarily interested the scenario where data acquisition costs are significantly more expensive than compute costs. This setup is common in the literature which deal with BC on human data (e.g. [1](https://arxiv.org/pdf/2108.03298) [2](https://arxiv.org/pdf/1805.01954)), where the model will be trained until convergence, for as many epochs as necessary.
>
> With that being said, we agree it is useful to report training-time behavior. We reran the experiments in section 4.1 to provide more clarity on the effect of skill on training time, and we present the results below. Overall, we see that the model trained on low-skill data not only has a higher task performance at low training budgets, but converges in fewer training epochs.
>
> | Budget  | Low-Skill Epochs |  High-Skill Epochs |
> | :-: | - | - |
> | 2^8 | 223.6 | 246.0 |
> | 2^10 | 161.9 | 189.2 |
> | 2^12 | 821.6 | 846.5 |
> | 2^14 | 1375.6 | 635.6 |
> | 2^16 | 1020.7 | 470.3 |
>
> > Composition vs. quantity: can you centralize, in the main text, the matched-budget low-only vs. high-only results and the fixed acquisition-budget schedule results to make the separation between “which data” and “how much data” unmistakable? Any sensitivity to alternative budget splits across skill bins?
>
> We agree that clearly distinguishing the effects of *composition* (“which data?”) from *quantity* (“how much data?”) is important for interpreting the results. Our intention in Sections 4 and 5 was precisely to separate these two perspectives, but we acknowledge that the distinction can be made more explicit in the main text.
>
> First, in the **matched-budget, low-only vs. high-only** experiments (e.g., Fig. 1 and Fig. 4), the total number of demonstrations is identical across skill levels. These experiments isolate the effect of **data composition alone** and show that low-skill datasets can outperform high-skill datasets under small budgets when their resilience is higher.
>
> Second, the **fixed acquisition-budget** experiments (e.g., the skill-based curricula in Fig. 3 and Fig. 5) represent the more realistic setting where data is pre-collected across skill bins, each contributing B/K demonstrations. These experiments address how to best use *mixed-skill* data once it has already been acquired, independent of the composition-vs.-quantity question.
>
> We will revise the paper to centralize this conceptual separation in the main text and clearly label which figures correspond to composition-only comparisons versus fixed-budget curriculum settings.
>
> Regarding sensitivity to alternative budget splits across skill bins, we note that the central finding—resilience predicting when low-skill data helps—does not depend on equal splits. In the tree domain, Fig. 7 already varies the underlying environment structure extensively, and in the chess domain (Fig. 8), intermediate skill levels naturally induce varied “effective” mixtures. Nonetheless, we agree that exploring alternative mixture ratios is an interesting extension, and we will note this as a direction for future work.

---

> ### Author Response · Authors · 2025-11-28
> **Response to Reviewer Y4Mt (3/3)**
>
> (continued from previous response)
>
> > Continuous control: for racing, how exactly are random actions sampled, and what proxies were used where direct resilience was infeasible? Please document the evaluator and sampling range.
>
> In racing, we could directly measure resilience by replacing a single action in the car's trajectory with a random action, where a random action is a tuple (t,s,b,g) sampled from $t \in [0,1], b \in [0,1], s \in [-2\pi, 2\pi], b \in [0,6]$ representing throttle, brake, steering, and gear. However, performing such perturbations at scale is currently impractical given computational costs and our access to the simulator, which limits us to running one model per physical machine/license in no faster than real-time. Instead, we demonstrate the difference in resilience between high-skill and low-skill models by leveraging randomness in the racing environment, e.g., due to bumps in the road, wind directions, slight variations in the driving line, etc. Appendix C.2 shows a real example of a low-skill model making a decision -- braking earlier before a turn -- that leads to higher resilience compared to a high-skill model which brakes too late. By braking earlier, the low-skill model enters states that can cope better with random perturbations such as bumps encountered during the actual turn.
>
> > Mixture selection: with a fixed budget and K skill bins, what mixed proportion of low/high skill maximizes performance? Even a small grid over mixtures in the synthetic domain would be informative.
>
> Thanks for the suggestion. We ran this experiment in the synthetic domain where we trained on mixed-skill datasets ranging from 100% low-skill to 100% high-skill. We found that at small budgets training on 100% low-skill data leads to higher performance than any introduction of high-skill data at small budgets. ([Figure](https://anonymous.4open.science/r/iclr2026-C020/tree/plots/mixtures.png))
>
> > Distribution shift: do the resilience performance links hold under test-time shifts (e.g., different chess openings or track conditions)?
>
> In the chess domain, we use a random opponent as an evaluation criteria, which is inherently already a test-time shift, as the random opponent tends to play very different openings than the original training dataset. In addition, we completed an additional set of evaluation results evaluating our models on a puzzle dataset (an even bigger distribution shift), and found the qualitative findings to stay the same. ([Figure](https://anonymous.4open.science/r/iclr2026-C020/chess/plots/puzzle-eval.png))

---

### Official Review · Reviewer_vLxH · 2025-11-03

**Soundness:** 2
**Presentation:** 2
**Contribution:** 1
**Rating:** 2
**Confidence:** 4

**Summary:**

Behavior cloning typically requires high-quality expert data, which restricts learning to such datasets. However, curating this data is difficult, whereas low-skill (worse-performing) data is more abundant. Is there a way we can use such data? This project explores this question with two main focuses: (1) what are the different qualities of high and low-skill data? (2) how can we use low-skill data in conjunction with high-skill data? To address (1), the authors define “resilence,” and show that for select datasets, low-skill data has higher resilience. Higher resilience means that the underlying policy generating the data is more robust to noise (e.g. the policy is able to recover after taking a noised action, whether this is due to a better policy overall, or the state coverage of the policy leading to less harmful/extreme states). To address (2), the authors explore curriculum using both high and low quality data. The authors show results on 3 domains: a synthetic toy experiment, chess, and racing.

**Strengths:**

1. The idea of resilience is interesting.
2. The intro and method section is fairly easy to read.

**Weaknesses:**

1. The experimental domains are a bit limited. The toy domain is useful for some analysis, but it is too synthetic. I’m not convinced by the chess or racing results.
2. The results aren’t too convincing. The gap between high and low skill is not too different.
3. Missing prior works in data quality and curation (DemoScore, DemInf, etc.)
4. The curriculum is not particularly novel.
5. Missing baseline where the policy is conditioned on the optimality of the data.

**Questions:**

1. I’m a bit confused by the training budget size. For example, for a given budget, if there are only two skill levels, the high-skill policy is trained on all high quality data, and the low-skill policy is trained on half high-quality and half low-quality, where the total amount of total is the same? (line 234)
2. How exactly is Eqn 1 calculated? Does the dataset include examples where different actions are taken from the same state? If so, I believe this is a strong assumption for many domains (although for chess, toy envs, etc., this is doable)
3. Could you calculate resilience with Monte Carlo rollouts in the environment, with added noise?
4. For Fig 4, is human data (fragile) a different opponent / environment altogether, or is it just a different training dataset.

---

> ### Author Response · Authors · 2025-11-28
> **Response to Reviewer vLxH (1/2)**
>
> We thank the reviewer for their thoughtful feedback. We are encouraged that they find the notion of resilience interesting. In our general response, we have provided details results in an additional experimental setting and provided theoretical insights for our resilience metric. Below we address each of the reviewers specific weaknesses/questions:
>
> > The experimental domains are a bit limited. The toy domain is useful for some analysis, but it is too synthetic. I'm not convinced by the chess or racing results.
> >
> > The results aren't too convincing. The gap between high and low skill is not too different.
>
> We appreciate the reviewer's concerns and would like to clarify why we believe the results are in fact strong and consistent across domains. First, as several other reviewers noted, the core phenomenon itself—that BC trained on *low-skill* data can outperform BC trained on *high-skill* data—is highly counterintuitive. Our experiments show this effect repeatedly and that the resilience ordering predicts it.
>
> Regarding the strength of the results: we would like to note that in both the tree domain (Fig. 1) and chess domain (Fig. 4), all curves include standard error bars. Since we conduct a large number of runs, the standard errors are very small that they are almost invisible, indicating highly stable outcomes over many random seeds. As a result, these differences are all statistically significant (p < 0.001). We also provide extensive sweeps—Fig. 7 (Tree) shows the phenomenon across a wide range of synthetic configurations, and Fig. 8 (Chess) extends the analysis across multiple intermediate skill levels (1200, 1600, 2000), with performance varying smoothly with skill and consistently aligning with the resilience ordering.
>
> For racing, due to restricted simulator access, the results are relatively limited, but the effect is nonetheless clear. For example, in Summit Point (Table 2), with a budget of 500 demonstrations, the low-skill model succeeds in 100% of trials while the high-skill model succeeds in 0%.
>
> Finally, in response to reviewer feedback, we have now replicated our main findings in MuJoCo and observe that resilience correctly predicts low-skill outperformance in 10 of 12 environments. We include a summary of these new results in the general response.
>
> > Missing prior works in data quality and curation (DemoScore, DemInf, etc.)
>
> We thank the reviewer for linking these works. We note in both works, the authors use their data curation approaches to filter out low-quality demonstrations during training. In contrast, in our approach, we show in Figure 3 and Figure 5 that we can improve on Filtered BC as a baseline, by leveraging low-quality demonstrations to improve the final model performance.  We will include discussion of these works.
>
> > The curriculum is not particularly novel.
>
> To clarify, the skill-based curriculum is not intended to be a core methodological contribution of the paper. Our main result is the counter-intuitive empirical finding that, in low-data regimes, BC trained on low-skill demonstrations can outperform BC trained on high-skill data, and that this effect is strongly predicted by our resilience metric.
>
> The curriculum experiments serve mainly to show that even a simple, off-the-shelf schedule, ordering data by demonstrator skill, can already make use of this insight and improve performance over standard or filtered BC. We fully agree that more principled curricula may provide further gains, but our goal here is simply to demonstrate that once resilience is understood, even a minimal curriculum can leverage it in practice.
>
> > Missing baseline where the policy is conditioned on the optimality of the data.
>
> We are not sure we understand the policy the reviewer is referring to. In Figure 1 (Tree), we have trained a model to mimic the optimal agent policy (the highest skill). In chess, no notion of optimality has been proven, but we have trained a model on the highest-skill agent currently available (Stockfish), and presented the results in Figure 4.

---

> ### Author Response · Authors · 2025-11-28
> **Response to Reviewer vLxH (2/2)**
>
> (continued from previous response)
>
> > I’m a bit confused by the training budget size. For example, for a given budget, if there are only two skill levels, the high-skill policy is trained on all high quality data, and the low-skill policy is trained on half high-quality and half low-quality, where the total amount of total is the same? (line 234)
>
> In the first set of results (Figures 1 and 4, Table 2), each treatment has the same budget size.
>
> In the second set of results (Figures 2 and 5), we have three treatments, which all consider a single mixed-skill dataset (e.g. 128 low-skill + 128 high-skill). In **Standard BC**, we train the model on the full dataset, all 256 demonstrations. In **Filtered BC**, we train on just the 128 highest-skill demonstrations. In **Filtered BC-LSC** (our contribution), we train in two stages - first on all 256 demonstrations until convergence, and then fine-tune on just the 128 highest-skill demonstrations until convergence.
>
> > How exactly is Eqn 1 calculated? Does the dataset include examples where different actions are taken from the same state? If so, I believe this is a strong assumption for many domains (although for chess, toy envs, etc., this is doable)
> >
> > Could you calculate resilience with Monte Carlo rollouts in the environment, with added noise?
>
> In general, we never expect the dataset to have multiple demonstrations at any state, and this is almost never true due to the large/continuous state spaces of our domains. Instead, we are assuming access to an approximate value function estimator which we can evaluate on states one step into the future (after making the random action). We then evaluate the resilience metric and average the result over all the states in the dataset.
>
> Resilience could be computed using a Monte-Carlo method. However, this shifts our requirements from having an approximate value function to having a strong policy function, which is probably a more difficult function to acquire, especially with only a limited amount of data, and the policy function must be held to a higher standard (more robust) than the value function, due to potential compounding errors.
>
> > For Fig 4, is human data (fragile) a different opponent / environment altogether, or is it just a different training dataset.
>
> In Figure 4, we have two sets of data sources. Our human data comes from lichess.org, and consists of games played between 800-rated players (low-skill), and games played between 2400-rated players (high-skill). Our Stockfish data consists of simulated games played between Stockfish at various depths to tune skill. When evaluating the models, we have the each trained model play against an opponent that plays entirely randomly.
>
> In response to reviewer feedback, we have also conducted a new set of evaluation experiments where the models play against an 800-rated BC model (trained to convergence on 256 million data points), and where the models solve various puzzles (a common evaluation metric in the Chess literature e.g. [1](https://arxiv.org/html/2402.04494v1) [2](https://arxiv.org/html/2409.12272v1)). In both setups, we corroborate our earlier findings to see that the model trained on low-skill data outperforms the high-skill model at low data budgets. ([Figure 1](https://anonymous.4open.science/r/iclr2026-C020/chess/plots/800-eval.png), [Figure 2](https://anonymous.4open.science/r/iclr2026-C020/chess/plots/puzzle-eval.png))

---

### Official Review · Reviewer_CDcV · 2025-11-06

**Soundness:** 2
**Presentation:** 3
**Contribution:** 2
**Rating:** 4
**Confidence:** 3

**Summary:**

This paper challenges the conventional wisdom in behavioral cloning (BC) that only expert demonstrations are useful. The authors show that in low-data regimes, training on low-skill demonstrations can outperform training on high-skill data when the low-skill policies are more resilient. They introduce a resilience metric to quantify this property and propose a skill-based curriculum (Filtered BC-LSC) that progressively incorporates higher-skill data. Experiments in synthetic tree environments, chess, and racing simulators validate their claims, demonstrating consistent improvements over standard and filtered BC baselines.

**Strengths:**

1. The authors provide a counterintuitive insight: Under certain conditions, specifically when the data-collecting policy exhibits higher resilience and the data budget is constrained, BC policies trained on low-skill data can outperform those trained on high-skill data.
2. Experimental results from multiple tasks across different domains demonstrate the validity of the theory.

**Weaknesses:**

1. The skill-based curriculum is heuristic and not theoretically justified; it would benefit from a more formal analysis or comparison to other curriculum strategies.
2. Lack of analysis on training time.

**Questions:**

1. In the evaluation of the chess task, the trained model is compared against a randomly-playing opponent. Is this a reasonable evaluation method? Can this approach genuinely reflect the performance quality of the trained policy?
2. Is the success rate in the racing experiment, obtained from only 5 attempts, statistically insufficient in terms of sample size?
2. Have you considered evaluating the approach in more complex or high-dimensional domains (e.g., robotics)?
3. Does the resilience of a policy relate to the diversity of states visited in low- vs. high-skill demonstrations?

---

> ### Author Response · Authors · 2025-11-28
> **Response to Reviewer CDcV (1/2)**
>
> We thank the reviewer for their thoughtful feedback. We are glad that they appreciate our counterintuitive central finding, and that they find the breadth of our results showing the validity of our approach.  In our general response, we have provided details results in an additional experimental setting and provided theoretical insights for our resilience metric. Below we address each of the reviewers specific weaknesses/questions:
>
> > The skill-based curriculum is heuristic and not theoretically justified; it would benefit from a more formal analysis or comparison to other curriculum strategies.
>
> To clarify, the skill-based curriculum is not intended to be a core methodological contribution of the paper. Our main result is the counter-intuitive empirical finding that, in low-data regimes, BC trained on low-skill demonstrations can outperform BC trained on high-skill data, and that this effect is strongly predicted by our resilience metric.
>
> The curriculum experiments serve mainly to show that even a simple, off-the-shelf schedule, ordering data by demonstrator skill, can already make use of this insight and improve performance over the standard methods used by practitioners (Standard and Filtered BC). We fully agree that more principled curricula may provide further gains, but our goal here is simply to demonstrate that once resilience is understood, even a minimal curriculum can leverage it in practice, and our findings pave the way for more advanced curricula to be developed.
>
> > Lack of analysis on training time.
>
> As noted on Line 235, we are primarily interested the scenario where data acquisition costs are significantly more expensive than compute costs. This setup is common in the literature which deal with BC on human data (e.g. [1](https://arxiv.org/pdf/2108.03298) [2](https://arxiv.org/pdf/1805.01954)). In this setup, the model will be trained until convergence, for as many epochs as necessary.
>
> With that being said, we agree it is useful to report training-time behavior. We reran the experiments in section 4.1 to provide more clarity on the effect of skill on training time, and we present the results below. Overall, we see that the model trained on low-skill data not only has a higher task performance at low training budgets, but converges in fewer training epochs.
>
> | Budget  | Low-Skill Epochs |  High-Skill Epochs |
> | :-: | - | - |
> | 2^8 | 223.6 | 246.0 |
> | 2^10 | 161.9 | 189.2 |
> | 2^12 | 821.6 | 846.5 |
> | 2^14 | 1375.6 | 635.6 |
> | 2^16 | 1020.7 | 470.3 |

---

> ### Author Response · Authors · 2025-11-28
> **Response to Reviewer CDcV (2/2)**
>
> (continued from previous response)
>
> > In the evaluation of the chess task, the trained model is compared against a randomly-playing opponent. Is this a reasonable evaluation method? Can this approach genuinely reflect the performance quality of the trained policy?
>
> Our choice of evaluating our models against a randomly playing opponent in chess was inspired by the self-play based approach of AlphaZero, which also evaluated each iteration of the model training cycle by playing against an equal or slightly weaker model. As our model is in the “first” iteration of self-play, being initialized randomly, we think it is natural to evaluate the model against a random opponent.
>
> To examine whether this choice impacts the finding, we re-ran our model evaluation against two additional baselines, against an 800-rated BC model (trained to convergence on 256 million data points) and evaluated the model's ability to solve chess puzzles (also a common evaluation metric in the Chess literature e.g. [1](https://arxiv.org/html/2402.04494v1) [2](https://arxiv.org/html/2409.12272v1)). In both setups, we corroborate our earlier findings to see that the model trained on low-skill data outperforms the high-skill model at low data budgets. ([Figure 1](https://anonymous.4open.science/r/iclr2026-C020/chess/plots/800-eval.png), [Figure 2](https://anonymous.4open.science/r/iclr2026-C020/chess/plots/puzzle-eval.png))
>
> > Is the success rate in the racing experiment, obtained from only 5 attempts, statistically insufficient in terms of sample size?
>
> Due to the limitations of our access to the training and simulator in racing, our results are relatively limited compared with our other domains, but it should hopefully still be evident that there is some effect happening. To answer your question about statistical significance, at the small budget range, when we look at our results over all tracks, the increase in performance between the low-skill (12/20) and high-skill models (4/20) is significant (p < 0.01). We will clarify these results in our writing. Additionally, to strengthen the generality of our findings, we have included results from a new domain as discussed in the next response.
>
> > Have you considered evaluating the approach in more complex or high-dimensional domains (e.g., robotics)?
>
> We thank the reviewer for this suggestion. We agree that a robotics domain would be valuable to demonstrate the efficacy of our approach. As a result, we have replicated our main set of results in MuJoCo, and found that resilience correlates with low-skill outperformance in 10/12 MuJoCo domains. Please see the general response to all reviewers for more details.
>
> > Does the resilience of a policy relate to the diversity of states visited in low- vs. high-skill demonstrations?
>
> The resilience measure is not directly correlated with state diversity. For instance, in the tree domain with fragile states, the low-skill policy is more resilient because it avoids brittle, high-penalty states and is less diverse. However, the high-skill policy actually visits more diverse states by deliberately entering critical states that low-skill agents avoid. Thus, higher state diversity does not imply higher resilience. We agree that exploring the relationship between resilience and state diversity is an interesting direction for future work.

---

### Author Response · Authors · 2025-11-28
**General Response to Reviewers (1/2) - Theoretical Insights on Resilience**

Thank you to all of the reviewers for their thoughtful feedbacks. We will be addressing each of the reviewers with individual responses to their questions shortly.

In the general response, we wanted to take this space to address two significant additions to our work which will be relevant to all reviewers. First, we will include additional theoretical justification for why resilience is a useful metric for predicting BC performance. Next, we will provide results in a set of 12 MuJoCo domains which align with both our previous experimental results and our new theoretical insights.

We have updated the manuscript with major changes highlighted in red text.

## Theoretical Insights on Resilience

First, we draw from the BC literature to explain the connection between resilience and BC performance, especially as it relates to differences in model performance across skills.

Let $\hat\pi$ be the BC policy trained on demonstrations from $\pi$, and let $\epsilon$ be its per-timestep imitation error (e.g., probability of deviating from the demonstrator action). Using Theorem 2.1 from [Ross 2011](https://www.ri.cmu.edu/pub_files/2011/4/Ross-AISTATS11-NoRegret.pdf) and adapting from a 0-1 loss formulation to a more general case, we obtain $J(\hat\pi) \approx J(\pi) - \epsilon T^2\ C_\pi$, where $C_\pi=J(\pi)-\mathrm{Res}(\pi)$ is the cost of making one error and following $\pi$ afterwards. This approximation assumes that BC errors follow a uniform distribution - that $\hat\pi$ takes the correct action with $1-\epsilon$ probability, and takes uniformly random actions with probability $\epsilon$, aligning with our definition of resilience. In practice, BC errors might not be uniform, and accounting for more realistic BC error in our resilience metric would be an interesting future direction. With this, we get:

$J(\hat{\pi}) \approx J(\pi)-\epsilon T^2 C_\pi$

$= J(\pi)-\epsilon T^2\big(J(\pi)-Res(\pi)\big)$

$= (1-\epsilon T^2)J(\pi)+\epsilon T^2 Res(\pi).$

This makes explicit that BC interpolates between demonstrator return $J(\pi)$ (when $\epsilon$ is small) and demonstrator resilience $Res(\pi)$ (when $\epsilon$ is larger). Let $\pi_{\text{low}}$ and $\pi_{\text{high}}$ be low- and high-skill demonstrators, with corresponding BC learners $\hat\pi_{\text{low}}, \hat\pi_{\text{high}}$. Applying the above approximation to each gives

$J(\hat\pi_{\text{low}}) - J(\hat\pi_{\text{high}}) \approx (1-\epsilon T^2)\big(J(\pi_{\text{low}})-J(\pi_{\text{high}})\big)+\epsilon T^2 \big(Res(\pi_{\text{low}})-Res(\pi_{\text{high}})\big)$

Empirically and theoretically, $\epsilon$ decreases as the demonstration budget increases. When $\epsilon$ is low (high data budgets), the skill term $\big(J(\pi_{\text{low}})-J(\pi_{\text{high}})\big)$ dominates, so training on $\pi_{\text{high}}$ yields better performance. **However, in the low data budget region when $\epsilon$ is large and $Res(\pi_{\text{low}}) > Res(\pi_{\text{high}})$, training on $\pi_{\text{low}}$ can yield better performance (i.e. $J(\hat\pi_{\text{low}}) > J(\hat\pi_{\text{high}}))$.**

---

> ### Author Response · Authors · 2025-11-28
> **General Response to Reviewers (2/2) - New experimental results in MuJoCo**
>
> (Continuing from above)
>
> ## New Experimental Results in MuJoCo
>
>
> Multiple reviewers suggested expanding our investigation to an additional domain. In response, we implemented the methodology described in Section 3.3.1 in MuJoCo and replicated our main finding: training on low-skill data can outperform training on high-skill data in low-budget regimes. We again observe a strong and consistent alignment between this phenomenon and our resilience metric. All code and results for the MuJoCo experiments are available at the [anonymized link](https://anonymous.4open.science/r/iclr2026-C020).
>
> **TLDR - We find that our proposed metric of resilience has a strong predictive power in MuJoCo. In five domains where low-skill agents are more resilient, models trained on low-skill trajectories outperform high-skill models in all 5 domains. In seven domains where high-skill agents are more resilient, high-skill models outperform in 5/7 domains. These results align with our existing domains and our new theoretical insights section.**
>
> | | Domain | Expert Resilience| Low-skill Resilience | Low-Skill is more resilient? | Low-Skill performs better at low budgets? | Aligns |
> |-|-|-|-|-|-|-|
> | Low-Skill Resilient | SAC-HalfCheetah | -0.32917 | **-0.211631** | ✅ | ✅ | ✅  |
> | | SAC-Hopper | 0.864271 | **0.906312** | ✅ | ✅ | ✅  |
> | | SAC-Humanoid | 4.807381 | **4.846941** | ✅ | ✅ | ✅  |
> | | TQC-HalfCheetah | -0.242331 | **-0.231321** | ✅ | ✅ | ✅  |
> | | TQC-Humanoid | 4.827656 | **4.848168** | ✅ | ✅ | ✅  |
> | High-Skill Resilient | PPO-Swimmer | **0.04955** | 0.044526 | ❌ | ❌ | ✅  |
> | | SAC-Ant | **-0.06562** | -0.345851 | ❌ | ❌ | ✅  |
> | | SAC-InvertedDoublePendulum | **7.490565** | 7.473262 | ❌ | ❌ | ✅  |
> | | SAC-InvertedPendulum | **0.678783** | 0.674888 | ❌ | ❌ | ✅  |
> | | SAC-Pusher | **-0.41236** | -0.420937 | ❌ | ✅ | ❌ |
> | | SAC-Reacher | **-0.14926** | -0.149267 | ❌ | ❌ | ✅  |
> | | SAC-Walker2d | **0.904928** | 0.812948 | ❌ | ✅ | ❌ |
>
> MuJoCo is a physics engine used for simulating robotics and biomechanics and contains 11 different environments to train agents for. We use an existing set of [pretrained models](https://huggingface.co/farama-minari/models) which span several combinations of environment, architecture, and agent skill. We took every environment/architecture combination with at least two models (except for SAC-HumanoidStandup, which failed to run), giving us 12 new domains to evaluate our results with.
>
> In each domain, we take the fully trained model (”Expert”), and compare it against the lowest-skill model available (”Simple” or “Medium”). For both models, we first compute the resilience score by rollout out the original policy for up to 1000 steps. Following Equation 3, we compute the counterfactual of the agent taking a random action at every step, and measuring the reward accumulated in the next 5 timesteps after this perturbation. We average these results over 1000 rollouts, and present the results in the above table. Overall, we see that in 5 domains, the low-skill agent demonstrates more resilience to random actions, and in the other 7 domains, the high-skill agent demonstrates more resilience.
>
> Next, we compute the task performance of models trained on low-skill data. Following the methodology from 3.3.1, we generate a dataset consisting of 1000-step rollouts by each agent we have access to. We then train models on these datasets, at budgets ranging from 2^2 to 2^10, and train these models until validation convergence. Finally, we evaluate the models by performing 10 rollouts, and reporting the cumulative reward attained. We repeat this procedure for 10 seeds, and present the results in the table above.
>
> In [this plot](https://anonymous.4open.science/r/iclr2026-C020/mujoco/plots/results.pdf), we report the results in all 12 domains. In the 5 environments with low-skill resilience, we find that all 5 plots, we find that low-skill models indeed outperform high-skill models consistently at low budgets. Additionally, in 5/7 domains with high-skill resilience, we find evidence that low-skill models consistently outperform high-skill models at low budgets.
>
> Overall, we find that the presence of low-skill resilience has 100% accuracy in predicting low-skill outperformance. When low-skill models are not more resilient though, low-skill models almost always fail to outperform high-skill models. The presence of the two exception domains (Reacher and Walker2d) does not invalidate our metric, but instead suggests that resilience does not completely explain the phenomenon of low-skill outperformance, and other factors (discussed in the general response section on theory), such as varying rates of $\epsilon$ across models or mistake-interactions may also play a part in this phenomenon.

---

### Author Response · Authors · 2025-12-03
**Summary Response for AC**

We are writing this note to summarize the full extent of the changes we have made to the paper during the rebuttal period to address the reviewers. We have responded to each of the reviewers in the comments, and have updated the pdf with major changes in highlighted in red text.

Overall, reviewers consistently found the paper novel, practically important, and conceptually intriguing, especially our concept of resilience and how it relates to the counterintuitive observation that low-skill demonstrations can sometimes outperform expert data in BC.

The two main suggestions consistently raised were:
- (1) Evaluation on additional domains (CDcV, vLxH, edUZ)
- (2) Clearer theoretical insight into the resilience metric (CDcV, Y4MT, edUZ)

We substantially strengthened the paper along both dimensions.

**Major Additions:**

- Evaluation on 12 new MuJoCo environments
  - We have evaluated our results on 12 new environments in MuJoCo, and these results are strongly consistent with our resilience metric predictions. In all 5/5 MuJoCo environments where our condition holds, the low-skill models outperform high-skill models. In 5/7 environment where our condition doesn't hold, the low-skill models do not outperform. These new domains directly address the reviewers’ request for additional evaluations and substantially broaden the empirical base of the paper. (Section 4.2)

- Theoretical insights on the resilience metric
  - We added a discussion connecting BC performance to our resilience metric through the literature on analyzing imitation learning error propagation. This provides intuitions on why and when training on low-skill data can surpass training on high-skill data. This responds directly to the reviewers’ request for theoretical insight. (Section 3.2)

In addition to these key points, we have added a number of clarifications and run new experiments to address the reviewer's other concerns and suggestions.

**Minor clarifications and additional experiments:**

- Clarification on error bars and statistical reporting (edUZ)
  - We clarified the reviewer confusion and stated that all plots in non-racing domains already include error bars, and added explicit statistical tests (p < 0.001 for tree and chess; p < 0.01 for racing). This resolves the reviewer confusion about missing statistical reporting. (Sections 4.1, 4.2, 4.4)
- Robustness to imperfect value functions (Y4MT)
  - We added a new chess experiment showing that the resilience ordering remains stable even when the value function is intentionally degraded. This directly strengthens the validity of the resilience metric and its use for predicting BC performance. (Table 5)
- Additional chess evaluation metrics (CDcV, vLxH, Y4MT)
  - We expanded the chess evaluation to include two new metrics, both of which confirm the original conclusions. This demonstrates that the phenomenon is not an artifact of a single evaluation protocol. (Tables 9, 10)
- Statistical significance in racing (CDcV, Y4MT)
  - We clarified that the racing results are statistically significant (p < 0.01), addressing concerns about reliability given the small number of rollouts. (Section 4.4)
- Training/compute time discussion + new experiment (CDcV, Y4MT).
  - We added discussion clarifying our focus on the common scenario where human demonstration acquisition is much more expensive than compute costs. That said, we also ran a new experiment showing that even if computing costs were a concern, low-skill models are cheaper to train and perform better at small budgets. (Table 4)
- Clarification on curriculum contributions (CDcV, vLxH, edUZ)
  - We clarified that the curriculum is not the central methodological contribution. We now explicitly separate the conceptual insight (resilience and its implications) from the curriculum schedule, which serves as one practical instantiation of the idea. (Section 1)

Overall, we added two major additions to our work during the rebuttal period (new theoretical section and 12 new MuJoCo domains), and addressed every single minor concern that the reviewers brought up.
As a result, we believe that this paper provides a high-value contribution to the literature, demonstrating to practitioners a theoretically sound, empirically validated approach to leveraging cheap, low-skill data to build more performant BC models.

---

### Meta-Review · Area_Chair_MBKZ · 2026-01-06

**Summary:**

This paper shows that in low-data regimes, training on low-skill data can outperform training on high-skill data when the low-skill policies are more resilient. It introduces a resilience metric to quantify the property and proposes a skill-based curriculum (Filtered BC-LSC) that progressively incorporates higher-skill data. The idea is interesting. Nevertheless, the reviewers have raised significant concerns, including: (1) limited experimental investigation (limited domains); (2) insufficient investigation over the resilence (narrow perturbation, lack of analysis, predictive capacity); and (3) lack of justification/formal analysis/comparisons over the curriculum learning scheme.

During the rebuttal, the authors provided additional results on MuJoCo tasks and offered theoretical insights on resilience, along with responses to several other issues. However, the repeated claim that the skill-based curriculum is not intended to be a core methodological contribution is unsatisfactory and ultimately weakens the paper’s overall contribution. The empirical finding that, in low-data regimes, behavior cloning (BC) trained on low-skill demonstrations can outperform BC trained on high-skill data is only meaningful if it is effectively incorporated into the training procedure, for example through a well-designed curriculum or another appropriate learning scheme. Overall, the paper received a low average rating of 3.5, which is insufficient to support acceptance.

**Reviewer Concerns:**

During the rebuttal, the authors provided additional results on MuJoCo tasks and offered theoretical insights on resilience, along with responses to several other issues. However, the repeated claim that the skill-based curriculum is not intended to be a core methodological contribution is unsatisfactory and ultimately weakens the paper’s overall contribution.

**Reviewer Scores:**

Given the reviewer scores (4, 2, 4, 4), substantial changes are unlikely.

---

### Decision · Program_Chairs · 2026-01-26

Reject